



# Methane retrieval from airborne HySpex observations in the short-wave infrared

Philipp Hochstaffl[1], Franz Schreier[1], Claas Henning Köhler[1], Andreas Baumgartner[1], and Daniele Cerra[1]

[1]Deutsches Zentrum für Luft- und Raumfahrt, Institut für Methodik der Fernerkundung, 82234 Oberpfaffenhofen, Germany

**Correspondence:** Philipp Hochstaffl (philipp.hochstaffl@dlr.de)

**Abstract.** A reduction of methane emissions could help to mitigate global warming on a relatively short time scale. Monitoring of local and regional anthropogenic $CH_4$ emissions is crucial in order to increase our understanding of the methane budget which is still subject to scientific debate.

The study compares various retrieval schemes that estimate localized $CH_4$ emissions from ventilation shafts in the Upper Silesian Coal Basin (USCB) in Poland using short-wave infrared nadir observations of the airborne imaging spectrometer HySpex. The examined methods are divided into nonlinear and linear schemes. The former class are of iterative nature and encompass various nonlinear least squares setups while the latter are represented by the Matched Filter (MF), Singular Value Decomposition (SVD) and Spectral Signature Detection (SSD) algorithms. Particular emphasis is put on strategies to remedy the problem of albedo related biases due to correlation with broad band absorption features caused by the hyperspectral 10 instrument's low spectral resolution.

It was found that classical nonlinear least squares fits based on the Beer InfraRed Retrieval Algorithm (BIRRA) suffers from surface-type dependent biases. The effect is more pronounced for retrievals from single spectral intervals but can be mitigated when multiple intervals are combined. The albedo related correlation is also found in the BIRRA solutions for the separable least squares. A new BIRRA setup that exploits the inverse of a scene's covariance structure to account for reflectivity statistics 15 significantly reduces the albedo bias and enhances the $CH_4$ signal so that the method infers two- to threefold higher methane concentrations.

The linear estimators turned out to be very fast and well suited to detect enhanced levels of methane. The linearized BIRRA forward model turned out to be sensitive to the selected retrieval interval and in the default setup only works for very narrow windows. Other well established linear methods such as the MF and SVD identified the methane pattern as well and largely 20 agree with the BIRRA fitted enhancements hence the methods allow quantitative estimates of methane. The latter two methods yielded increased performance when the scene was further divided into clusters by applying k-means in a preprocessing step. Methane plumes detected with the simple SSD method were faint and found rather sensitive to the polynomial used to compute the method's residuum ratio.


## 1  Introduction

Methane ($CH_4$) is the second most important greenhouse gas next to carbon dioxide ($CO_2$) according to the latest IPCC report (Masson-Delmotte et al., 2021). Due to its comparatively short lifetime of approximately 9 years, a reduction of methane emissions could help to mitigate global warming on a relatively short time scale of approximately one decade. Despite improvements in monitoring regional and global $CH_4$ emissions in recent years the IPCC report points out that fundamental uncertainties pertaining to the methane budget remain (Intergovernmental Panel on Climate Change, 2014).

Observations indicate an increasing trend in atmospheric $CH_4$ content since 2007, the cause of which is still subject to scientific debate. The vast majority of anthropogenic $CH_4$ emissions is caused by small scale phenomena such as agriculture (enteric fermentation & manure), waste management (landfills) and fossil fuel exploitation, where the latter is responsible for 20-30 % of all anthropogenic $CH_4$ emissions. Consequently there exists the need for continuous long-term methane observations on a global scale, in order to foster understanding on the global methane cycle, devise future reduction measures and

monitor their effectiveness. The monitoring of anthropogenic emissions of $CH_4$ and $CO_2$ is also part of the United Nations Framework Convention on Climate (2015) as nationally determined contributions should be assessed via global stock takes on a 5 year basis from 2023 (Article 13 & 14 of the Paris Agreement).

Satellite observations are typically the method of choice for such continuous and global long-term observations. Space-borne spectrometers measuring short-wave infrared (SWIR) solar radiation reflected at the Earth surface are especially well suited to

observe atmospheric $CH_4$ in the lower atmosphere by measuring its absorption in the bands 1560-1660 nm and 2090-2290 nm. In contrast, the thermal infrared is less sensitive to variations in $CH_4$ concentration close to the surface. Moreover, thermal sensors often have lower spatial resolution making them less favorable for emission monitoring (Richter, 2010).

Operational $CH_4$ products from contemporary atmospheric composition missions such as TROPOMI (TROPOspheric Monitoring Instrument; Veefkind et al. (2012)), GOSAT/GOSAT-2 (Greenhouse gases Observing SATellite; Kuze et al. (2009, 2016))

measure trace gas concentrations with very high accuracy, nevertheless, they are not optimally suited to measure emissions of point-like sources. This design inherent limitation is due to their focus on rapid global coverage, which entails a comparatively coarse spatial resolution of several square kilometers per pixel. Since the emission of a single point source inside a pixel is averaged over the entire resolution cell, even large sources seldomly elevate the mean $CH_4$ concentration within one pixel by more than one percent compared to the undisturbed background (Lauvaux et al., 2022). A way to increase the contrast

of enhancements is to operate typical atmospheric remote sensing spectrometers at lower altitudes (e.g. on aircraft), thus increasing the spatial resolution while leaving the overall optical design untouched. This strategy is followed by instruments such as MAMAP/MAMAP-2D (Gerilowski et al., 2011) or GHOST (Humpage et al., 2018) which are very well suited for the calibration and validation of their space-borne counterparts.

In order to increase the sensitivity towards smaller sources an increased spatial resolution is required, which in turn ne-

cessitates a trade-off in spectral resolution because the loss of photons caused by the smaller emitting area per pixel reduces the Signal-to-Noise Ratio (SNR) of the image which has to be compensated by broadening the spectral interval per spectral channel. Imaging spectrometers for land surface remote sensing (often referred to as hyperspectral cameras) are typical exam-



ples of instruments optimized for spatial resolution this way. Their technology matured over the last 30 years and a variety of airborne instruments and several space-borne versions are either in orbit (PRISMA, Guanter et al., 2021; ENMAP, Chabrillat

et al., 2020) or going to be launched in the future (CHIME). Yet other sensors dedicated for the detection of methane (GHGSat, Jervis et al., 2021) and carbon dioxide (e. g., Carbon Mapper, CO2Image) have slightly higher spectral resolution than their hyperspectral counterparts but still offer a much higher spatial resolution than atmospheric composition missions.

Thorpe et al. (2013) were the first to demonstrate that localized $CH_4$ emissions over land can be detected from hyperspectral cameras with the Airborne Visible/Infrared Imaging Spectrometer (AVIRIS, Green et al. 1998) and that a limited quantitative

analysis is possible (Thorpe et al., 2014). Similar studies were repeated with airborne instruments (AVIRIS-NG, Borchardt et al. 2021; HySpex, Nesme et al. 2020) and space-borne instruments (Thompson et al., 2016; Guanter et al., 2021). Works by Varon et al. (2019); Jervis et al. (2021) demonstrated that $CH_4$ sources can even be detected with the multi-spectral MSI instrument on-board the Sentinel-2 satellites, but these measurements are restricted to 'favourable conditions' (i. e., strong sources and high surface albedo).

One of the core challenges when retrieving methane from measurements with high spatial and moderate spectral resolution ($> 1$ nm) is the separation of spectral variations caused by molecular absorption and surface reflectivity. Classical trace gas retrievals for high-spectral resolution instruments such as RemoteC (Lorente et al., 2021), Weighting Function Modified Differential Optical Absorption Spectroscopy (WFM-DOAS, Buchwitz et al., 2005), or the Beer InfraRed Retrieval Algorithm (BIRRA, Gimeno García et al., 2011) exploit the high frequency characteristics of gaseous absorption and attribute the smooth

varying part to the surface albedo. Instruments with coarse spectral resolution, however, are unable to sufficiently resolve those molecular signatures which causes ambiguities that often leads to surface-type related biases in the 'classical' retrieval schemes (e. g., Borchardt et al., 2021, Sec. 3.3 or Thorpe et al., 2014, Sec. 9.2). Alternative more 'data-driven' retrieval schemes such as the Matched Filter (MF) or the Singular Value Decomposition (SVD) estimate enhancements based on methods from linear algebra and statistics (Thorpe et al., 2013; Thompson et al., 2015; Thorpe et al., 2014).

This study compares various retrieval schemes applied to measurements from DLR's (German Aerospace Centre) HySpex sensor system and the paper is structured as follows. The next section briefly describes the experimental setup, provides a quick review of atmospheric radiation, and introduces the various BIRRA setups that were examined in this study. Thereafter, other (simpler but faster) retrieval schemes employed in this work are briefly described. The result section starts with a feasibility analysis for BIRRA with (simulated) HySpex data and proceeds with the presentation of the retrieval results from HySpex

observations for different BIRRA setups over the Pniowek V ventilation shaft. Thereafter, the results from the well established 'data driven' fitting techniques such as MF and SVD, are presented. In the last chapter, results are sumarized and put into perspective.

## 2 Methodology

Methods introduced in this section can be divided into linear and nonlinear schemes. While the former are very fast and often

of sufficient accuracy the nonlinear iterative solvers require more computing power and hence time to come up with a best





estimate. The retrieval methods are taylored to remedy the problem of albedo related biases due to correlation with broad band absorption features caused by the instrument's low spectral resolution.

## 2.1 Experimental Setup

The measured spectra analyzed in the study at hand were acquired during the COMET (Carbon diOxide and METhane) campaign with the DLR HySpex sensor system. This airborne imaging spectrometer, which consists of two commercially available hyperspectral cameras (a VNIR-1600 and a SWIR-320m-e) is described in detail in (IMF) and references therein.

(a)

(b)

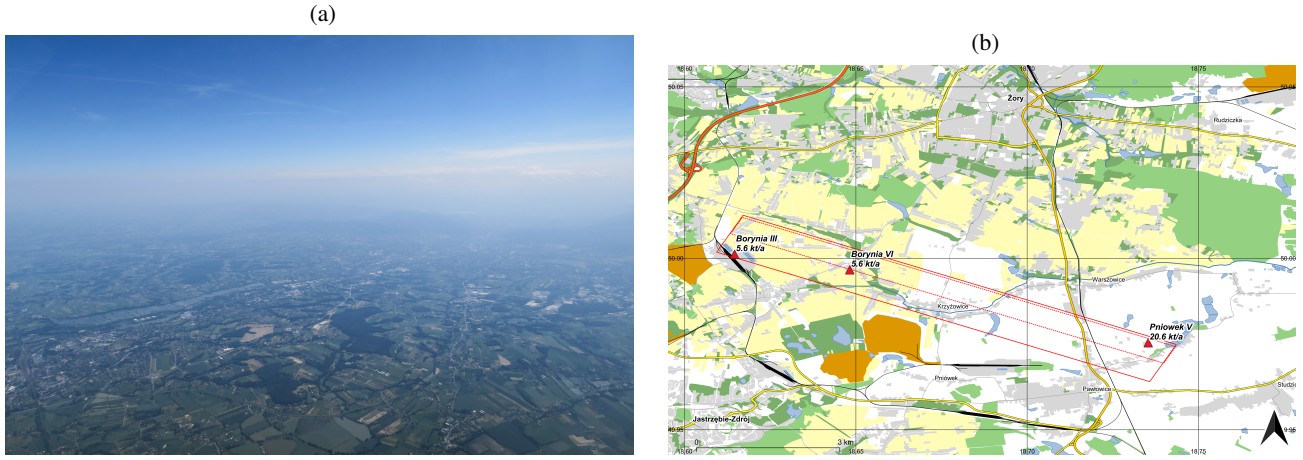

**Figure 1. (a)** View from the aircraft into the mountains around Zywiec at 10 UTC on June 07, 2018. **(b)** In the depicted flight track ("scene 09") the aircraft was on a 115 degrees eastbound heading at $\approx 1.5$ km above mean sea level. The map was created with QGIS using OpenStreetMap data (© OpenStreetMap contributors, 2022).

The data analyzed in the following chapters was collected during a survey flight conducted within the scope of the CoMet campaign on June 7th, 2018. The CoMet campaign focused on the detection and characterization of $CO_2$ and $CH_4$ sources in the Upper Silesian Coal Basin in southern Poland. It featured a number of ground-based and airborne measurements with both in-situ and remote sensing instruments. The HySpex survey was intended as a feasibility study to evaluate whether – and if so how accurate – localized methane emissions can be retrieved from the SWIR-320m-e data. To achieve this goal we planned 18 flight lines at two different altitudes over a number of known ventilation shafts around Katowice. The location and estimated emission rate of the ventilation shafts was taken from the CoMet ED v1 inventory assembled by Nickl et al. (2020). It was not known in advance, though, which of these ventilation shafts would be actively emitting methane during the day of the survey, as the emission rates are derived from monthly averages reported by the mining companies operating the shafts. The weather during the survey was well suited for remote sensing measurements. Apart from very few occasional patches of thin cirrus clouds there were no further low or mid-level clouds. However, a significant amount of haze could be observed from the aircraft. This can be seen in Image 1a, which displays a view from the aircraft towards the mountains around Bielsko-Biala,





located approximately 20 km southeast of the survey area. Actual wind data for the USCB area on the measurement day is
presented in Luther et al. (2022, Fig. 4 and 6).

Since this study compares the performance of various retrieval methods, we restrict our analysis to the two flight lines: Flight
line 9, acquired at 1200 m above ground level (AGL) around 0955 UTC and flight line 11 acquired at 2600 m AGL around
1010 UTC. The respective foot prints of line 9 (dashed red line) and line 11 (solid red line) are shown in Fig. 1b along with the
location and estimated emission rate of the three ventilation shafts (red triangles) located within. Each track took the aircaft
115   $\approx$ 3 minutes during which 7130 (scene 09) and 5075 (scene 11) along track observations for each of the 320 across track
detector pixels were acquired.

In Fig. 2b an ensemble of along track averaged HySpex measurements are depicted. The sensor's sampling distance across
the spectral axis is indicated by the vertical grid lines which is not constant in the wavenumber domain. The spectral coverage
of the HySpex SWIR-320m-e camera ranges from 967–2496 nm (4005–10338 cm$^{-1}$), with the exact number depending on the
120   across track pixel ($\approx \pm 1$ cm$^{-1}$). The spectral resolution, i. e., the full width at half maximum (FWHM) of the HySpex SWIR-
320m-e camera in the 4000–6500 cm$^{-1}$ region ranges from 6.0–9.5 nm (10–40 cm$^{-1}$). Its values are provided for each across
track pixel of the detector (a 2D array) with the level 1b data set. This data set was basically created as described in Lenhard
et al. (2015), except for the optical distortion correction. The Instrument Spectral Response Function (ISRF) calibration was
performed according to Baumgartner (2021). Hence, the ISRF for each pixel is available as a lookup table with an sampling
125   distance of 1.2 nm. The standard HySpex product is corrected for optical distortions and resampled to a constant spectral
resolution and across-track resolution using the method described in Baumgartner and Köhler (2020). For this study, this
processing step was ommited.





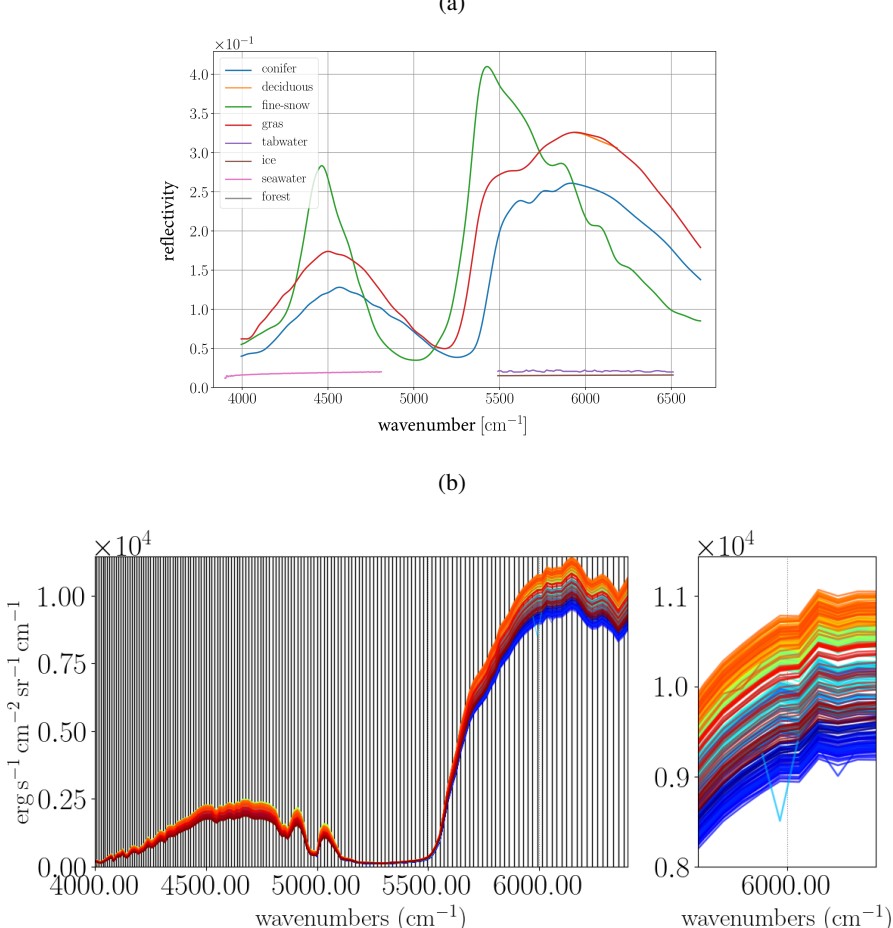

**Figure 2. (a)** Reference reflectances for different surface types (measured at the John Hopkins University). **(b)** HySpex measurements across the 320 detector pixels (from blue to red, left to right). The radiance values of across track pixel 104 (cyan) for wavenumber $5960\,\mathrm{cm}^{-1}$ (relevant for the CH4 retrieval) appear to be problematic.

As pointed out, it is the rather low spectral resolution that makes the retrieval of atmospheric constituents challenging. Figure 2 shows reflectances for various surface types along track averaged Hyspex spectra. Note that the radiative intensity in the interval around $6000\,\mathrm{cm}^{-1}$ is significantly larger compared to the radiance between $4000$–$5000\,\mathrm{cm}^{-1}$. The measurement is only able to resolve broad band molecular absorption features since the high frequency variations are smoothed by the coarse instrument resolution (see absorption from methane's $2\nu_3$ band around $6000\,\mathrm{cm}^{-1}$). The figure also indicates a possible bad pixel with systematically lower radiance values along the flight track, just below $6000\,\mathrm{cm}^{-1}$, corresponding to across track pixel 104 (a descending cyan line).



## 2.2 Radiative transfer

In the SWIR spectral range the radiative transfer for a down and up path through the atmosphere under clear sky conditions (cloud free) is well described by Beer's law (Zdunkowski et al., 2007) with the monochromatic transmission from Top of Atmosphere (TOA) to Bottom of Atmosphere (BOA) given by

$$\mathcal{T}_m(\nu;s) \;=\; \exp\left(-\sum_m \tau_m(\nu,s)\,\mathrm{d}s\right) \;=\; \exp\left(-\int\limits_{\mathrm{path}} \mathrm{d}s \sum_m n_m(s)\,k_m\big(\nu,p(s),T(s)\big)\right). \tag{1}$$

The model assumes a pure gas atmosphere with molecular optical depth $\tau$ given by the path integral over the molecular number densities $n_m$ and $k_m$, the pressure and temperature dependent absorption cross section.

In conditions where particles such as haze, dust or high clouds prevail, extinction (scattering and absorption) by aerosols should be taken into account (De Leeuw et al., 2011). Aerosol optical thickness $\tau_{\mathrm{aer}}$ at wavenumber $\nu$ is often described by a power law

$$\tau_{\mathrm{aer}}(\nu) \;=\; \tau_{\mathrm{aer}}(\nu_0)\left(\frac{\nu}{\nu_0}\right)^{\beta}, \tag{2}$$

where $\tau_{\mathrm{aer}}(\nu_0)$ is the optical thickness at a reference wavenumber and $\beta$ a parameter for the aerosol. The Ångstrom exponent $\beta$ typically ranges from $1 \leq \beta \leq 2$ (Liou, 2002) and while it is close to $1.0$ for almost clear sky conditions with weak scattering by haze or dust, it is assumed to increase for hazy conditions. In analogy to $k_m$ the aerosol cross section can be defined as

$$\tau_{\mathrm{aer}}(\nu_0) \;=\; \int\limits_{\mathrm{path}} k_{\mathrm{aer}}(\lambda_0)\,n_{\mathrm{air}}(s)\,\mathrm{d}s \;=\; N_{\mathrm{air}}\,k_{\mathrm{aer}}(\lambda_0)\,\big(10^4/\nu\big)^{-\beta} \tag{3}$$

with $\lambda_0 = 1\,\mu\mathrm{m}$ and $k_{\mathrm{aer}}(\nu)$ proportional to $\lambda^{-\beta}$ according to

$$k_{\mathrm{aer}}(\lambda) \;=\; k_{\mathrm{aer}}(\lambda_0)/\lambda^{\beta} \quad\text{and}\quad k_{\mathrm{aer}}(\lambda_0) \;=\; 1.4\cdot 10^{-27}\,. \tag{4}$$

## 2.3 Model atmosphere setup

The model atmosphere's vertical extent ranges from $0$–$80\,\mathrm{km}$ with 39 levels in total. The atmosphere is composed by pure gaseous layers above $z_{\mathrm{mol}} = 10\,\mathrm{km}$ and layers containing gases and particles below $z_{\mathrm{sc}} = 10\,\mathrm{km}$. The vertical resolution is highest in the (plume) layer below $z_{\mathrm{pl}} = 2\,\mathrm{km}$ where the enhancement is expected to takes place (see Fig. 3). The $CH_4$ optical depth is modeled in terms of a climatological background and a Gaussian plume

$$\tau_{CH_4} \;=\; \tau_{\mathrm{bg}} \,+\, \alpha_{CH_4}\,\tau_{\mathrm{pl}}\,. \tag{5}$$

Although the shape of the plume profile is not crucial as the nadir viewing geometry does not allow to retrieve information on the vertical distribution of trace gases in the SWIR (see Buchwitz et al. (2000, Sec. 3)) our setup constrains the fit to the lowest atmospheric layer up to $2.0\,\mathrm{km}$ (see Thorpe et al. (2014, 5.2)).





The $CH_4$ background as well as the $CO_2$ initial guesses are modeled according to the Air Force Geophysical Laboratory (AFGL, Anderson et al., 1986) atmospheric constituent profiles scaled to 1875 ppb and 400 ppm, respectively. The molecules $H_2O$ as well as the auxiliary parameters temperature and pressure are taken from reanalysis data provided by the National Center for Environmental Prediction (NCEP, Kalnay et al., 1996).

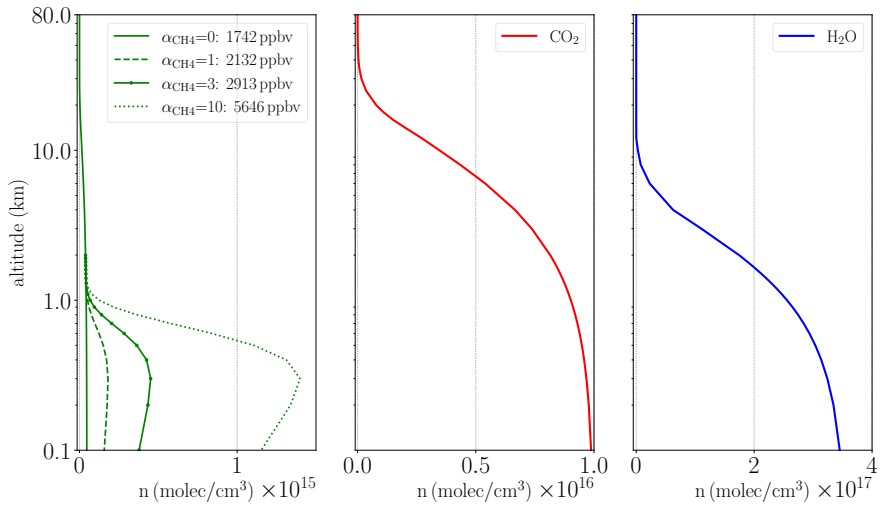

**Figure 3.** Atmospheric vertical profiles of molecular number densities $n_m$ for $CH_4$, $CO_2$, and $H_2O$. The $CH_4$ profile is split into two components, i. e., a reference (background) profile ranging from BoA to ToA and a Gaussian plume enhancement $< 2\,km$ which is scaled by $\alpha_{CH4}$. Beside $CH_4$ the well mixed $CO_2$ profile is depicted in the middle while an initial guess for $H_2O$ is shown on the right.

## 2.4 Beer InfraRed Retrieval Algorithm (BIRRA)

The BIRRA level 2 processor was originally developed at the Deutsches Zentrum für Luft- und Raumfahrt (DLR) and comprises the line-by-line forward model Generic Atmospheric Radiation Line-by-line InfraRed Code (GARLIC, Schreier et al., 2014) coupled to a least squares solver for trace gas retrieval in the SWIR spectral region (Hochstaffl et al., 2018). It has been successfully applied to SCIAMACHY (Scanning Imaging Absorption Spectrometer for Atmospheric Chartography, Gimeno García et al., 2011; Hochstaffl and Schreier, 2020) and TROPOMI (TROPOspheric Monitoring Instrument, Hochstaffl et al., 2020) observations. The BIRRA retrievals in this study are based on a Python reimplementation of the validated (Fortran) code (Gimeno García et al., 2011; Hochstaffl et al., 2018). The radiative transfer computations are hence based on Py4CAtS (Python for Computational Atmospheric Spectroscopy, (Schreier et al., 2019)), a Python reimplementation of GARLIC.

The mathematical forward model $\Phi(\boldsymbol{x},\nu)$ describes the measured intensity spectrum $I(\nu)$ for a nadir looking observer according to

$$\Phi(\boldsymbol{x},\nu) \;=\; \frac{r(\nu)}{\pi}\,\cos(\theta)\,I_{\mathrm{sun}}(\nu)\,\mathcal{T}_m^{\downarrow}(\nu)\,\mathcal{T}_m^{\uparrow}(\nu)\otimes S(\gamma(\nu))\,, \tag{6}$$





where $r$ refers to the surface reflectivity and $\theta$ represents the solar zenith angle. The terms $\mathcal{T}_m^{\downarrow}$ and $\mathcal{T}_m^{\uparrow}$ denote the total transmission between Sun and reflection point (e.g. the Earth) and between reflection point and observer (e.g. the HySpex sensor), respectively. The transmission by aerosols for different Ångstrom exponents according to Eq. (2) is depicted in Fig. 4 (center).

180  Its behavior can be represented by a low order polynomial hence the forward's model total transmission is described as

$$\mathcal{T}_m(\nu; s) = \exp\left(-\sum_m \alpha_m \, \tau_m(\nu) - \sum_{i>0} a_i \, \nu^i\right). \tag{7}$$

The unknown (to be estimated) parameters are composed as elements of the state vector $x$ and include the molecular scaling factors $\alpha_m$, the aerosol coefficients $a_i$, and the coefficients for the surface reflectivity $r_j$ (with $j \geq 0$) which is also modeled by a polynomial. Note that since the information of the vertical profile is well under-determined in the observed spectrum scaling

185  factors $\alpha_m$ for the initial guess profiles are retrieved (Gimeno García et al., 2011, Fig. 1). Finally, the instrument's spectral response is described by the spectral response function (SRF) $S$. Its parameters such as the half width $\gamma$ or a spectral shift can (optionally) be part of the state vector.



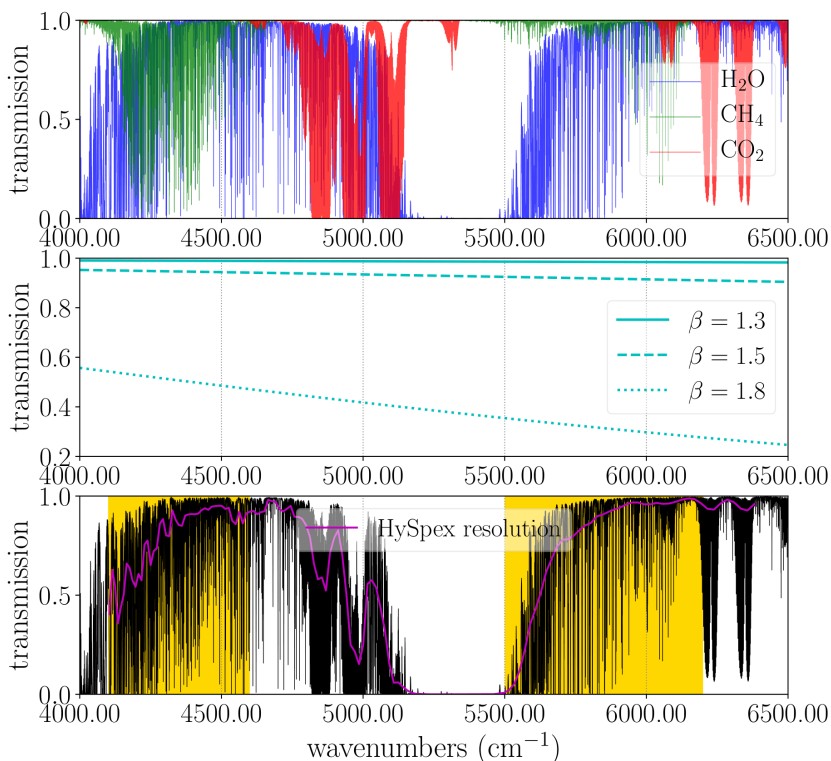

**Figure 4.** Monochromatic transmissions of $CH_4$, $CO_2$ and $H_2O$ for the SWIR spectral range and a nadir looking observer at 1.5 km at a solar zenith angle (SZA) of $30°$ are depicted in the top panel. The aerosol transmission in the middle panel shows only smooth variations across the spectrum. The magenta line in the lower panel represents the total transmission degraded to HySpex resolution and the spectral intervals with methane absorption are indicated by the yellow background. Also note significant differences in transmissions of the monochromatic spectrum and convolved instrument spectrum.

The molecular absorption calculations in this study exploit GEISA (Gestion et Etude des Informations Spectroscopiques Atmosphériques; Delahaye et al., 2021) 2020 spectroscopic line data. In the top panel of Fig. 4 the individual components of the monochromatic total transmission for the US-Standard atmosphere are shown. Methane's first overtone of the fundamental vibrational transition $2\nu_3$ (with its P and R branches) is found around $6000\,\mathrm{cm}^{-1}$ while additional (strong) absorption lines range from $4100$–$4700\,\mathrm{cm}^{-1}$ (band center $\approx 4420\,\mathrm{cm}^{-1}$). The panel at the bottom demonstrates how the total monochromatic transmission (in black) is smoothed by the observer's coarse spectral resolution.

In Fig. 5 the BIRRA Jacobian matrix for two spectral intervals with strong methane absorption is depicted, respectively.





(a)                                                          (b)

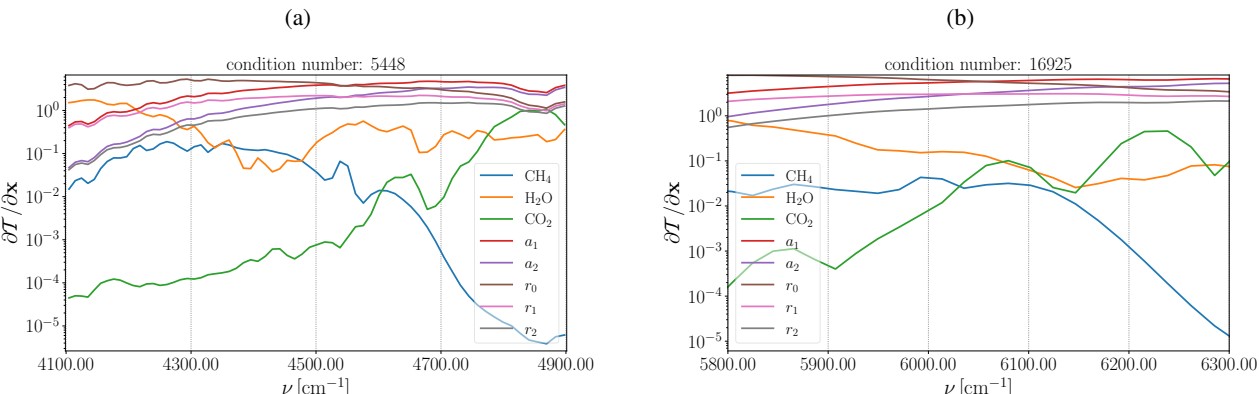

**Figure 5.** Columns of the Jacobian matrix and condition numbers in the $4100$–$4900\,\mathrm{cm}^{-1}$ (left) and $5800$–$6300\,\mathrm{cm}^{-1}$ (right) spectral intervals.

### 2.4.1 Nonlinear least squares solvers

Various nonlinear retrieval schemes were examined and are briefly introduced subsequently. The iterative nature of nonlinear least squares methods requires the calculation of derivatives for each of the nonlinear state vector elements across the spectral axis which is represented by the Jacobian matrix $\mathsf{J}$. The feasibility of a given retrieval setup is briefly studied after the introduction of the various solvers in Sec. 3.1.1. Therefore the condition number of the Jacobians are examined for different spectral intervals relevant for the SWIR $CH_4$ retrieval.

The retrieval's performance, e. g., the fit quality, is assessed with respect to the 2-norm of the discrepancy between the measurement $\boldsymbol{y}$ and the converged spectrum $\sigma = ||\boldsymbol{y} - \boldsymbol{I}(\boldsymbol{x})||_2$, also known as the residual norm. The least squares (error) covariance matrix is given by

$$\Xi = \frac{\sigma^2}{(m-n)}\left(\mathsf{J}^\mathrm{T}\,\mathsf{J}\right)^{-1} \tag{8}$$

where $\mathsf{J}$ represents the Jacobi matrix, while $m$ and $n$ specify the number of measurements and number of state vector elements, respectively. The errors of the individual state vector parameters are obtained from the diagonal elements of $\Xi$.

**Nonlinear least squares (NLS)**

The nonlinear least squares fit minimizes the objective function $\mathcal{L}$ for given measurements $\boldsymbol{y}$ according to

$$\min_x \left\{\mathcal{L}(\boldsymbol{x})\right\} = \min_{\boldsymbol{x}} ||\boldsymbol{y} - \boldsymbol{\Phi}(\boldsymbol{x})||_2^2 , \tag{9}$$

and applies when the model function $\boldsymbol{\Phi}$ is nonlinear in one or more parameters of $\boldsymbol{x}$.



**Separable least squares (SLS)**

The so called separable least squares solver splits (separates) the state vector $\boldsymbol{x}$ into nonlinear and linear parameters $\boldsymbol{x} = (\boldsymbol{\eta}, \boldsymbol{\beta})$ where the elements in $\boldsymbol{\beta}$ enter the forward model $\boldsymbol{\Phi}$ linearly (see Sec. 2.5) while the components in $\boldsymbol{\eta}$ are of nonlinear nature. The minimization problem is hence given by

$$\min_{\eta,\beta} \|\boldsymbol{y} - \boldsymbol{\Phi}(\boldsymbol{\eta})\,\boldsymbol{\beta}(\eta)\|_2^2 \, . \tag{10}$$

This setup is also known as the Variable Projection (VarPro, Golub and Pereyra, 2003) method where $\boldsymbol{\eta}$ is independent of $\boldsymbol{\beta}$ in the matrix product $\boldsymbol{\Phi}(\boldsymbol{\eta})\,\boldsymbol{\beta}(\eta)$. The parameters in $\eta$ can hence be fitted in the usual way by means of Gauss–Newton or Levenberg–Marquardt algorithms (see e. g. Hansen et al. (2013)).

**Generalized least squares (GLS)**

A so called generalized least squares fit can be employed to account for correlated errors. The covariance matrix $\mathsf{C}$ is used to account for the spectral variations of the scene's background, i. e.parts of the flight track which are presumably not influenced by the $CH_4$ plume. Therefore, the location of the point source along with wind data needs to be knows. The matrix $\mathsf{C}$ is then created by computing the spectral covariance for a given scene. The idea is that possible background variations similar to the methane band absorption are not (mistakenly) interpreted as a molecular enhancement. The covariance matrices for the methane retrieval intervals are depicted in Fig. 6.

The symmetric positive semidefinite error covariance matrix $\mathsf{C}$ is (pre-)computed for a given flight track so that the non-negative square root matrix $\mathsf{S} = \mathsf{C}^{\frac{1}{2}}$ can be used to estimate $\boldsymbol{x}$ according to

$$\min_{\boldsymbol{x}} \|\mathsf{S}^{-1}(\hat{\boldsymbol{y}} - \mathsf{J}\boldsymbol{x})\|_2^2 \, . \tag{11}$$





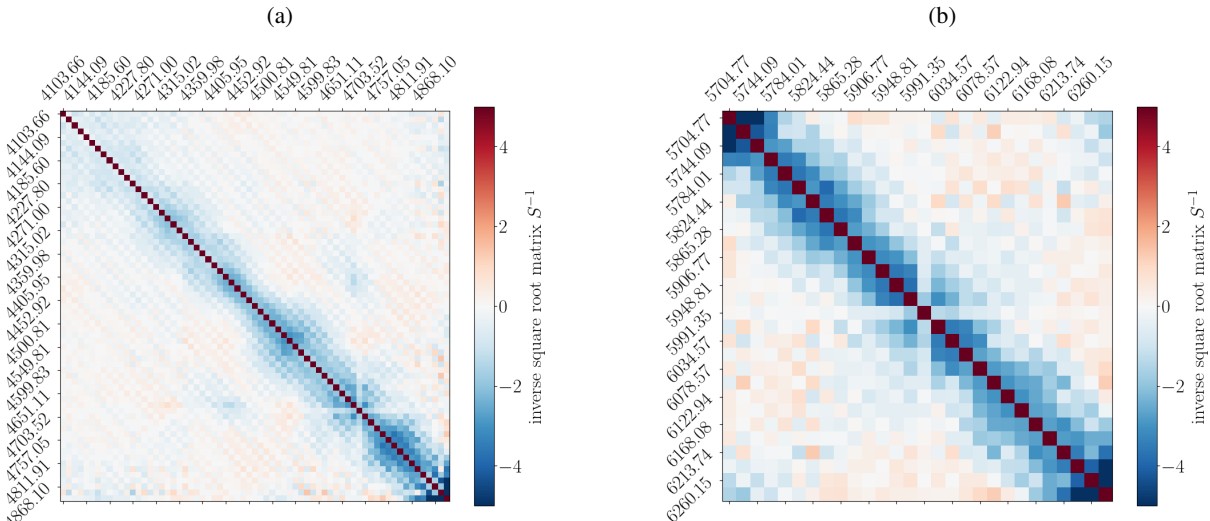

**Figure 6.** Scene 09 background covariance matrix $\mathsf{S}$ for the **(a)** 4100-4900 $\mathrm{cm}^{-1}$ (left) and **(b)** 5700-6300 $\mathrm{cm}^{-1}$ (right) spectral range. Note that beside the bad HySpex pixel mentioned in Fig. 2 at 5992.74 $\mathrm{cm}^{-1}$ there appears to be another suspect pixel at 4691.04 $\mathrm{cm}^{-1}$.

### 2.4.2 Methane enhancement estimate

In order to estimate the $CH_4$ plume enhancement light path modifications and the retrieval's vertical sensitivity need to be taken into account. A 'scene average' $CO_2$ scaling factor is used to correct for light path modifications (Frankenberg et al., 2005; Schneising et al., 2009; Krings et al., 2011; Borchardt et al., 2021). In order to apply the $\alpha_{CO2}$ scaling factor to $CH_4$ enhancements for the correction of light path modifications below instrument altitude, the different concentration profiles of $CH_4$ and $CO_2$ (see Fig. 3) need to be taken into account hence $\tilde{\alpha}_{CO2}$ is introduced. The 'scene averaging' method was also

applied to infer the actual $CH_4$ background profile for the respective overpass. It is important to note that for the $CO_2$ and $CH_4$ background fits ground pixels around the suspected $CH_4$ sources were excluded when computing the average spectrum. The lack of vertical atmospheric resolution in the observed spectrum requires the application of averaging kernels in order to account for the retrieval's altitude sensitivity. The column averaging kernel $\kappa_m(z)$ is used to describe the sensitivity of the total columns to changes in molecular concentrations at different levels (see Buchwitz et al., 2004).

The actual $CH_4$ column, which includes corrections for light path modifications via $\tilde{\alpha}_{CO2}$ and accounts for the retrieval's vertical sensitivity with respect to the target by $\kappa_{CH4}(z)$, is finally computed as the sum of the background and plume component according to

$$N_{CH4} = N_{bg} + \frac{\alpha_{CH4}}{\tilde{\alpha}_{CO2}} \hat{N}_{pl}(z_0) \,, \tag{12}$$





with

$$\hat{N}_{\mathrm{pl}}(z_0) = \int\limits_{z_0}^{z_{\mathrm{pl}}} \frac{n_{\mathrm{pl}}(z)}{\kappa_{\mathrm{CH4}}(z)}\,\mathrm{d}z\,. \tag{13}$$

The (highly variable) water vapor concentration is co-retrieved with the $CH_4$ plume enhancements as results indicate degeneracy between $H_2O$ and the reflectivity polynomial so that the $H_2O$ scaling factor and reflectivity coefficients need to be interpreted as 'effective' parameters that capture low frequency components in the spectrum. The 'mixing' of usually clearly separated spectral features is attributed to the coarse spectral resolution of HySpex measurements and the fact that water vapor

absorption lines of similar strength are distributed over a wide spectral range.

### 2.5 Linear fitting algorithms

**Linear least squares (LLS)**

Linearization of the BIRRA forward model with respect to $\alpha_{\mathrm{CH_4}}$ allows to infer methane enhancements by linear least squares. In analogy to Thompson et al. (2015, Sec. 2.4) where the $CH_4$ enhancement is estimated from the linear scaling of a target

signature that perturbs the mean radiance, linearization of Beer's law caused by an increase in methane's total optical depth in the lowest part of the atmosphere ($< 2\,\mathrm{km}$) with respect to the (saturated, see Thompson et al. (2015)) background concentration is justified.

In order to estimate the unknown parameters in $x$ by linear least squares the power-series expansion for the exponential function $\exp(\tau) := \sum_{n=0}^{\infty} \frac{\tau^n}{n!}$ is exploited. Assuming that the increased optical depth caused by the plume $\tau_{\mathrm{plume}}$ is rather

small the Taylor expanded transmission spectrum for the plume can be approximated as

$$\exp\left(-\tau_{\mathrm{plume}}^{\downarrow\uparrow}(\nu)\right) \approx \left(1 - \beta_{\mathrm{CH4}}\,\tau_{\mathrm{plume}}^{\downarrow\uparrow}(\nu)\right)\,. \tag{14}$$

The forward model for the linear least squares problem of $M$ measurements can then be formulated according to

$$\{\boldsymbol{\Phi}(\boldsymbol{x})\}_i = \sum_{j=1}^{N} x_j\,\phi_j(\nu_i), \qquad i = 1, 2, \cdots, M \tag{15}$$

so that the model functions for the linear parameters of the state vector $\boldsymbol{x} = (r_0, b_0 = r_0\,\beta_{\mathrm{CH4}})$ are given by

$$\phi_1 = \frac{\cos(\theta)}{\pi} I_{\mathrm{sun}}\,\mathcal{T}^{\downarrow\uparrow} \otimes S\,,$$

$$\phi_2 = -\frac{\cos(\theta)}{\pi} I_{\mathrm{sun}}\,\mathcal{T}^{\downarrow\uparrow}\,\tau_{\mathrm{plume}}^{\downarrow\uparrow} \otimes S\,.$$

Note that the reflectivity coefficient $r_0$ is present in both elements of $\boldsymbol{x}$. However, this should not pose a problem for the linear fit as the model functions are different. A brief analysis on the condition of $\boldsymbol{\Phi}(\boldsymbol{x})$ in the 5700–6300 $\mathrm{cm}^{-1}$ interval revealed a condition number is 885. When the higher order reflectivity coefficient $r_1$ is included the number increases by one order of

magnitude and another order if $r_2$ added. Therefore, in the current setup, the linear fit is only feasible for the estimate of $r_0$ and $\beta_{\mathrm{CH4}}$ at the same time. However, using standardized radiances by dividing by a fitted polynomial eliminates the need for higher order reflectivity coefficients even for large intervals.


### 2.5.1 Matched Filter (MF)

A well established method to estimate molecular concentration enhancements from hyperspectral sensors is the MF (Theiler
and Foy, 2006; Villeneuve et al., 1999; Funk et al., 2001; Thompson et al., 2015). The linear enhancement factor estimate is
based on the perturbation of an average (background) radiance spectrum $\boldsymbol{\mu}$ by a known target spectrum $\mathbf{t}$ and is formulated
according to

$$\beta(\boldsymbol{y}) \;=\; \frac{(\mathsf{J} - \boldsymbol{\mu})^{\mathrm{T}}\,\mathsf{C}^{-1}\,(\boldsymbol{y} - \boldsymbol{\mu})}{\sqrt{(\mathsf{J} - \boldsymbol{\mu})^{\mathrm{T}}\,\mathsf{C}^{-1}\,(\mathsf{J} - \boldsymbol{\mu})}} \,. \tag{16}$$

The method tests an observed vector $\boldsymbol{y}$ against a base vector represented by e. g. the $CH_4$ plume Jacobian $\mathsf{J}$ (computed with a
radiative transfer model, e. g., Py4CAtS) while accounting for the background covariance $\mathsf{C}$. The method also assumes that the
measured spectrum can be represented as a linear superposition of the plumes optical depth and the mean radiance $\boldsymbol{\mu}$ according
to

$$\boldsymbol{y} \;\approx\; \boldsymbol{I} \;=\; \boldsymbol{\mu}(1 - \tau_{\mathrm{pl}}^{\downarrow\uparrow}\,\beta)\,. \tag{17}$$

In order to allow for a comparison to the BIRRA setups, the target's signature $\tau_{\mathrm{pl}}^{\downarrow\uparrow})$ represents the vector of optical depth for a
low level plume ($< 2\,\mathrm{km}$ with $390\,\mathrm{ppm}\ CH_4$) that is scaled by the linear enhancement factor $\beta$ that perturbs the mean radiance.
Note that the mean background spectrum $\boldsymbol{\mu}$ and $\mathsf{C}$ were computed per scene and the inverse covariance $\mathsf{C}^{-1}$ is approximated
by decomposing $\mathsf{C}$ into eigenvalues and eigenvectors (Thompson et al., 2015, Eq. 6-8).

    As pointed out by Guanter et al. (2021) the classical matched filter is relatively sensitive to surface albedo hence also the
cluster tuned matched filter (Funk et al., 2001) was examined. Classification of the image reduces the within-class variance
which in turn should reduce the albedo sensitivity of $\beta$. So instead of computing a single covariance (background) statistic
the cluster tuned matched filter computes background statistics $\mathsf{C}_i$ for each cluster $i$ determined by k-means (Thorpe et al.,
2013; Nesme et al., 2020). The elbow method (Thorndike, 1953) was employed to estimate the suitable number of clusters for
a scene.

### 2.5.2 Singular Value Decomposition (SVD)

The retrieval of methane enhancements from hyperspectral AVIRIS data using singular vectors of the observed spectrum plus
a target signature was first demonstrated by Thorpe et al. (2014). In this study the uncorrelated (orthogonal) singular vectors
are obtained from HySpex spectra within a scene (flight track) that are (assumed to be) not impacted by the plume. The scene's
mean standardized background spectrum was decomposed while the target spectrum represented by the $CH_4$ plume's Jacobian
given by

$$\boldsymbol{t} \;=\; -\boldsymbol{\mu}\exp\left(-\tau_{\mathrm{CH4}}\right)\tau_{\mathrm{pl}} \qquad \text{with} \qquad \tau_{\mathrm{CH4}} \;=\; \tau_{\mathrm{bg}} + \beta\,\tau_{\mathrm{pl}} \tag{18}$$

was computed with the radiative trasfer code Py4CAtS. Note that $\tau_{\mathrm{bg}}$ represents methane's background optical depth for the
lowest, plume impacted layers.





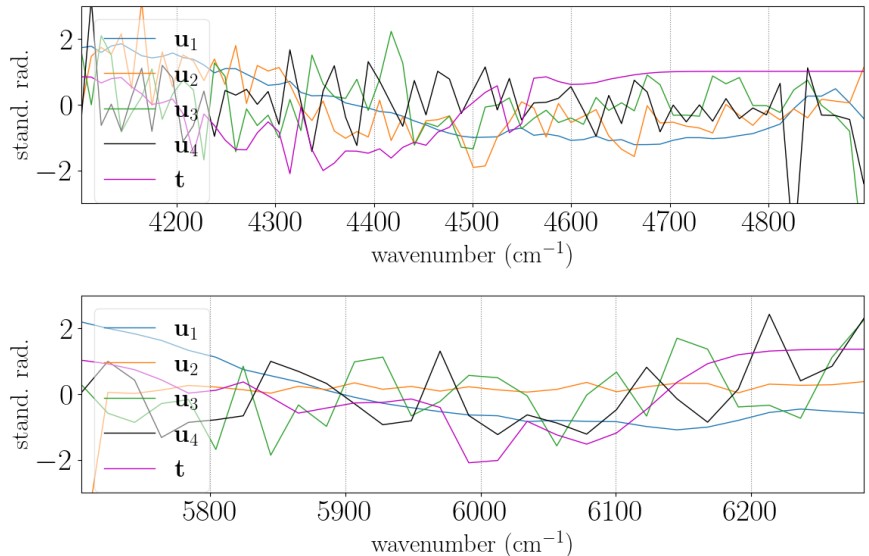

**Figure 7.** The first four singular vectors after standardization, i. e., removing the mean and scaling to unit variance, along with the methane plume's target signature $\mathbf{t}$.

The basic idea corresponds to the MF, i. e., to represent the general variability in spectral radiance by a linear combination of singular vectors and a target signal. The minimization problem is then given by

$$\min_{\boldsymbol{w}} \|\boldsymbol{y} - \mathsf{A}\,\boldsymbol{w}\|_2^2 \quad \text{with} \tag{19}$$

$$\mathsf{A}\,\boldsymbol{w} = \sum_{k}^{N} \boldsymbol{u}_k\,w_k + \boldsymbol{t}\,w_{\text{CH4}} \tag{20}$$

where $\mathsf{A}$ represents the concatenated matrix of the first $N$ columns of the unitary matrix $\mathsf{U}$ and the vector $\boldsymbol{w}$ contains the corresponding weights. The contribution of enhanced methane in the lowest atmospheric layers to the measured radiance is estimated by the corresponding weight of the $\text{CH}_4$ target signature.

It was found that $N = 4$ is a good choice across spectral intervals as including additional columns significantly increased the condition number of $\mathsf{A}$. It should be noted that cluster tuning was also examined for the SVD fits. In this case the background spectra were clustered by k-means and the SVD performed for each cluster separately and the respective base vectors per cluster were then used in the linear fit.

### 2.5.3 Spectral signature detection (SSD)

A very simple yet effective method to detect elevated concentrations of methane in a flight track (subsequently designated as a 'scene') is based on the ratio of spectral residual norms. This method does not require any radiative transfer calculations, look up tables or initial guess information but only calibrated (digital numbers are sufficient) sensor data for a given interval.



The algorithm is based on a simple polynomial fit of spectral pixels and the computation of spectral residuals. The idea of the method is similar to the continuum interpolated band ratio (CIBR) Green et al. (1989) and Thompson et al. (2015, Eq. 2) which also scores absorption depths (Pandya et al., 2021).

The application of our signature detection method requires the spectral interval including $m$ pixels to be separated into section(s) where $CH_4$ absorbs and where it does not (or only weakly). The detection method then applies a linear least squares to fit a polynomial of some degree $P$ to $Q$ out-band pixels

$$p(\boldsymbol{\beta}, \nu_i) = \sum_{j=0}^{P} \beta_j \, \nu_i^j, \qquad i = 1, 2, \cdots, Q. \tag{21}$$

Next the residual norms

$$r = \|\boldsymbol{y} - \boldsymbol{p}\|_2^2. \tag{22}$$

for the out-band pixels and $(M - Q)$ in-band-pixels are computed so that their ratio

$$s = \frac{r_{\mathrm{in}}}{r_{\mathrm{out}}} \tag{23}$$

yields an absorption band depth score for each observation. Variations in the score indicate variations in the $CH_4$ absorption. If a constant (zero order polynomial) is used for the out-band fit, the method is similar to the CIBR algorithm. With higher order polynomials the wavenumber dependent contributions from the surface reflectivity and other known interfering species could be modeled more accurately, particularly over larger intervals.

The algorithm constitutes a fast detection scheme which can also be applied for real-time detection of enhancements, e. g., determine whether or not a $CH_4$ ventilation shaft is active at the time of instrument overpass.

## 3 Results

In this section the results of the prescribed retrieval methods are presented based on HySpex nadir observations in the short-wave infrared. The section starts with a feasibility study for different BIRRA setups. The results focus on HySpex measurements from flight track 9 (scene 9) since it was found to be the one with the strongest emission at the time of overpass so that the results of the various retrieval schemes can be compared. Note that this ventilation shaft was actually overpassed two times, i. e., in flight tracks 9 and 11 from aircraft altitude $\approx 1520\,\mathrm{m}$ and $\approx 2900\,\mathrm{m}$, respectively. This circumstance could e. g. be used to study the impact of spatial resolution on concentration enhancements.

### 3.1 Beer InfraRed Retrieval Algorithm

The section begins with a feasibility assessment for different BIRRA state vectors and presents the retrieval results from HySpex measurements using the retrieal's NLS, SLS and GLS setups.





### 3.1.1 Feasibility of BIRRA state vectors

According to Table 1 the reduced parameter space in the separated least squares improves the condition number of its Jacobian. The assessment also revealed that increasing the spectral resolution by a factor of two improves the condition number by $\approx 10\,\%$ meaning that the condition number in Fig. 5 would be $\approx 30\,\%$ lower if HySpex would measure at a resolution of $0.2$–$0.3\,\mathrm{cm}^{-1}$ FWHM.

**Table 1.** Condition numbers for the Jacobian matrices of various state vectors of the intervals 4100–4900 $\mathrm{cm}^{-1}$ (designated as 4K) and 5700–6300$\,\mathrm{cm}^{-1}$ (6K), respectively. The upper part of the table shows the condition numbers for the nonlinear fit while the lower part gives the conditions for the Jacobians only containing the nonlinear parameters (required for the VarPro solver). The state vector component $N_m = 3$ represent the three molecular scaling factors ($\alpha_{\mathrm{CH4}}$, $\alpha_{\mathrm{H2O}}$, $\alpha_{\mathrm{CO2}}$), $N_a$ stands for the aerosol scaling factors (e. g., $a_1$, $a_2$), and $N_r$ represents the number of coefficients (e. g., $r_0, r_1, r_2$) of the reflectivity polynomial.

| $x$ | 4100–4900 $\mathrm{cm}^{-1}$ (4K) | 5700–6300$\,\mathrm{cm}^{-1}$ (6K) | combined |
|---|---|---|---|
| $N_m = 3, N_r = 3, N_a = 0$ | 189 | 1323 | 147 |
| $N_m = 3, N_r = 3, N_a = 1$ | 730 | 2269 | 187 |
| $N_m = 3, N_r = 3, N_a = 2$ | 6681 | 50625 | 198 |
| $\eta$ | 4100–4900$\,\mathrm{cm}^{-1}$ (4K) | 5700–6300$\,\mathrm{cm}^{-1}$ (6K) | - |
| $N_m = 3$ | 9 | 32 | - |
| $N_m = 3, N_a = 1$ | 47 | 499 | - |
| $N_m = 3, N_a = 2$ | 88 | 839 | - |

### 3.1.2 Nonlinear least squares

The state vector $x = (3m, 3r)$ was found to be robust toward low SNR values across the examined spectral intervals and is hence the first choice for the subsequent retrievals. However, because $CO_2$ is required to account for light path modifications, the actual retrieval fits the state vector $x = (\alpha_{\mathrm{CH4}}, \alpha_{\mathrm{H2O}}, 3r)$. The scene averaged $CO_2$ background level is inferred from the $1.6\,\mu\mathrm{m}$ and/or $2\,\mu\mathrm{m}$ bands via the multi-interval fit. The decision to exclude aerosol parameters from the $CH_4$ plume fit was also encouraged by findings from Borchardt et al. (2021), as they conclude that different aerosol scenarios in the SWIR do not induce errors $> 0.2\,\%$. Moreover, since the spectra are observed at low flight altitudes (between $1500\,\mathrm{m}$ and $3000\,\mathrm{m}$ above mean sea level (MSL)) on a clear day (Luther et al., 2019) retrieval errors induced by aerosol scattering should be negligible in our scenario too (also see Fig. 4) (Thorpe et al., 2013; Thompson et al., 2015). Nonetheless, aerosol extinction according to Eq. (2) could still be included as a given input argument in the forward model.

The $CO_2$ was found to be 385–400 ppm for scene 09 and 365–380 ppm for scene 11 depending on the spectral interval. Given twice the instrument altitude for scene 11 the relative difference of the retrieved values for both scenes is reasonable as changes in $CO_2$ are attributed to light path modifications (light path shortening due to single scattering). The average difference of 20 ppm motivated the choice of a scaling factor $\tilde{\alpha}_{\mathrm{CO2}} = 0.96$ for scene 09 and $\tilde{\alpha}_{\mathrm{CO2}} = 0.93$ for scene 11. The scene averaged



CH$_4$ background profile was also determined in advance and found to be within 5 % of the initial guess of 1850 ppb so that its
initial guess was not scaled.

In Figs. 8a and 8b the results for the retrieval from $3 \times 3$ averaged observations for scene 09 from 4150–4900 cm$^{-1}$ (4K)
and 5700–6300 cm$^{-1}$ (6K) are shown, respectively. Averaging over multiple HySpex observations is a way to increase SNR
and reduces scattering of the inferred quantities. In both spectral intervals a CH$_4$ enhancement is identified and both reveal
significant levels of CH$_4$ (up to $\approx$4000 ppb) at and close to the source (ventilation shaft). Furthermore, the results agree on
the direction of advection. Interestingly the surface-type dependent bias is opposite in the 4K and 6K intervals. The issue of
different surfaces types and their impact on the uncertainty of CH$_4$ quantification for moderately resolved spectra was also
described by Borchardt et al. (2021) who observed similar features, i. e., paved concrete induces a positive bias while barbed
goatgrass leads to large underestimation of enhancements (and so the total column) as at rather coarse spectral resolutions the
reflected spectrum shows interfering features similar to the absorption of CH$_4$.





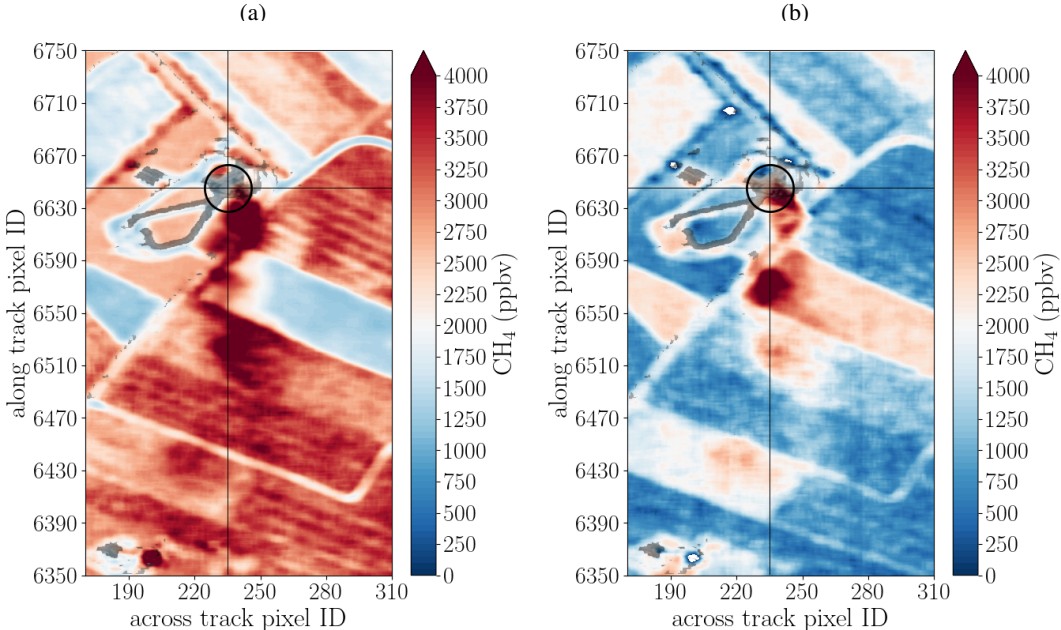

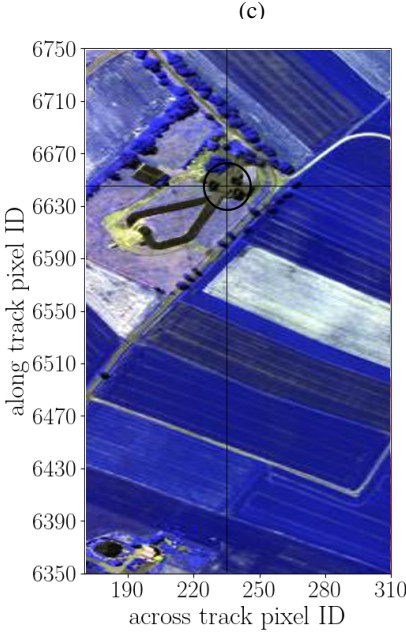

**Figure 8.** Retrieved $CH_4$ enhancement of $3 \times 3$ spatially averaged HySpex observations in the **(a)** 4150–4900 cm$^{-1}$ (4K) range and **(b)** 5700–6300 cm$^{-1}$ (6K) range. Interestingly, the albedo related biases show an almost identical pattern but reverse sign. **(c)** False color image of the SWIR-320m-e camera around the Pniovek V shaft in scene 09.





In spite of a significantly stronger signal in the 6K range (see Fig. 2), the significantly higher condition number of its Jacobians (see Table 5) make both intervals similarly suitable for the retrieval of methane.

With this finding the multi-interval retrieval, i. e., combining 4K and 6K ranges, is expected to yield better results given that methane is fitted across intervals so that the additional constraint alleviates albedo induced variabilities on the target. Figure 9 depicts the inferred $CH_4$ enhancements for scene 09 with the multi-interval retrieval. The surface correlated bias is still present

but reduced compared to the single-interval fits. The maximum enhancements and pattern of the $CH_4$ plume is similar but the downwind shape of the plume is better captured.

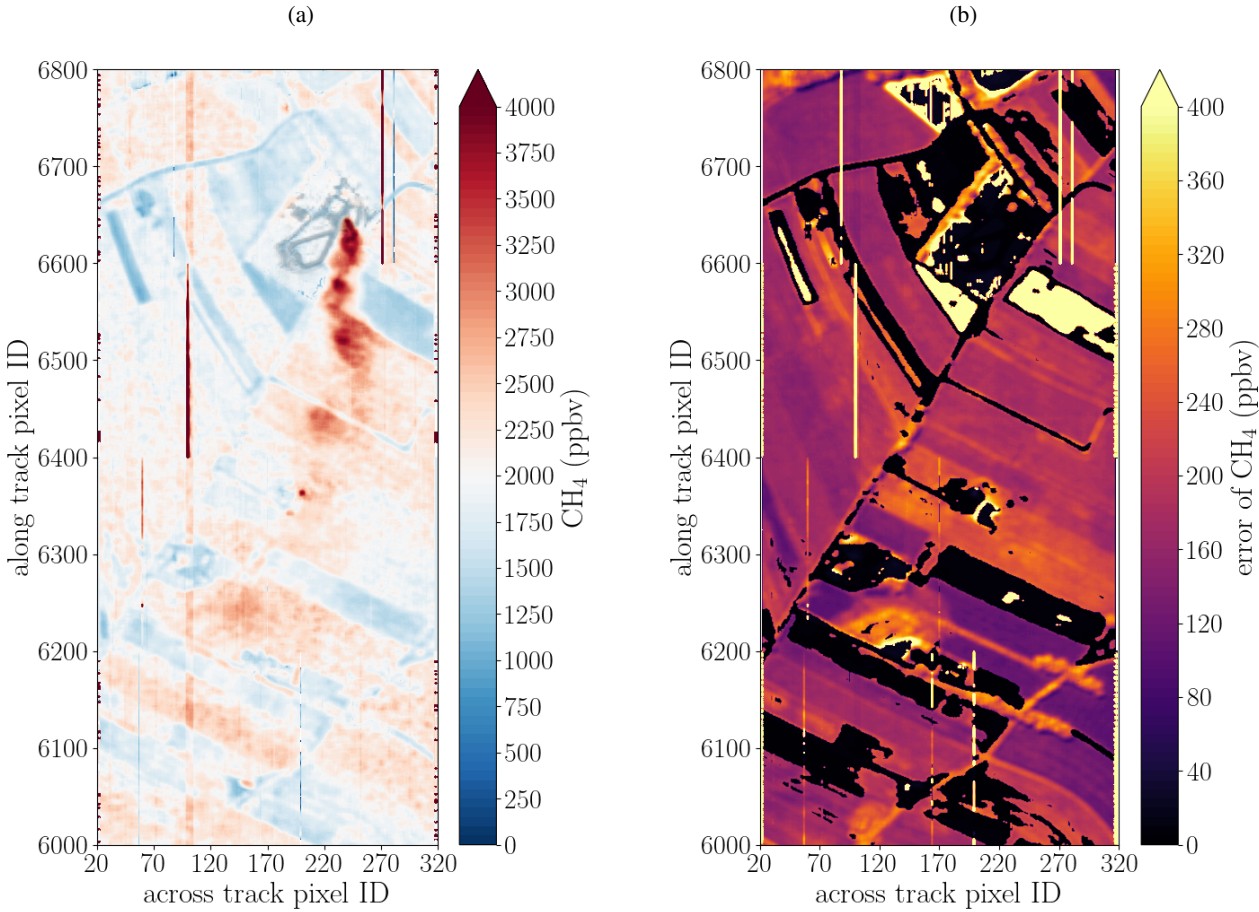

**Figure 9. (a)** Methane plume with corresponding errors in **(b)** inferred for $3 \times 3$ spatially averaged HySpex observations with the multi-interval fit for the combined intervals 4150–4900 cm$^{-1}$ and 5700–6300 cm$^{-1}$. The fitted state vector was $\boldsymbol{x} = (\alpha_{CH4}, \alpha_{H2O}, 6r)$. The bluish colors correlate with either high or very low errors indicating observations with either a small albedo or reflectivity which could not be captured by the second order polynomial.





In another setup that in addition includes the aerosol parameter $a_1$ similar $CH_4$ concentrations were inferred. However, the $a_1$ estimate adversely affects the fit of the reflectivity coefficients. The impact on $r_0(6K)$ was found to be stronger than on $r_0(4K)$.

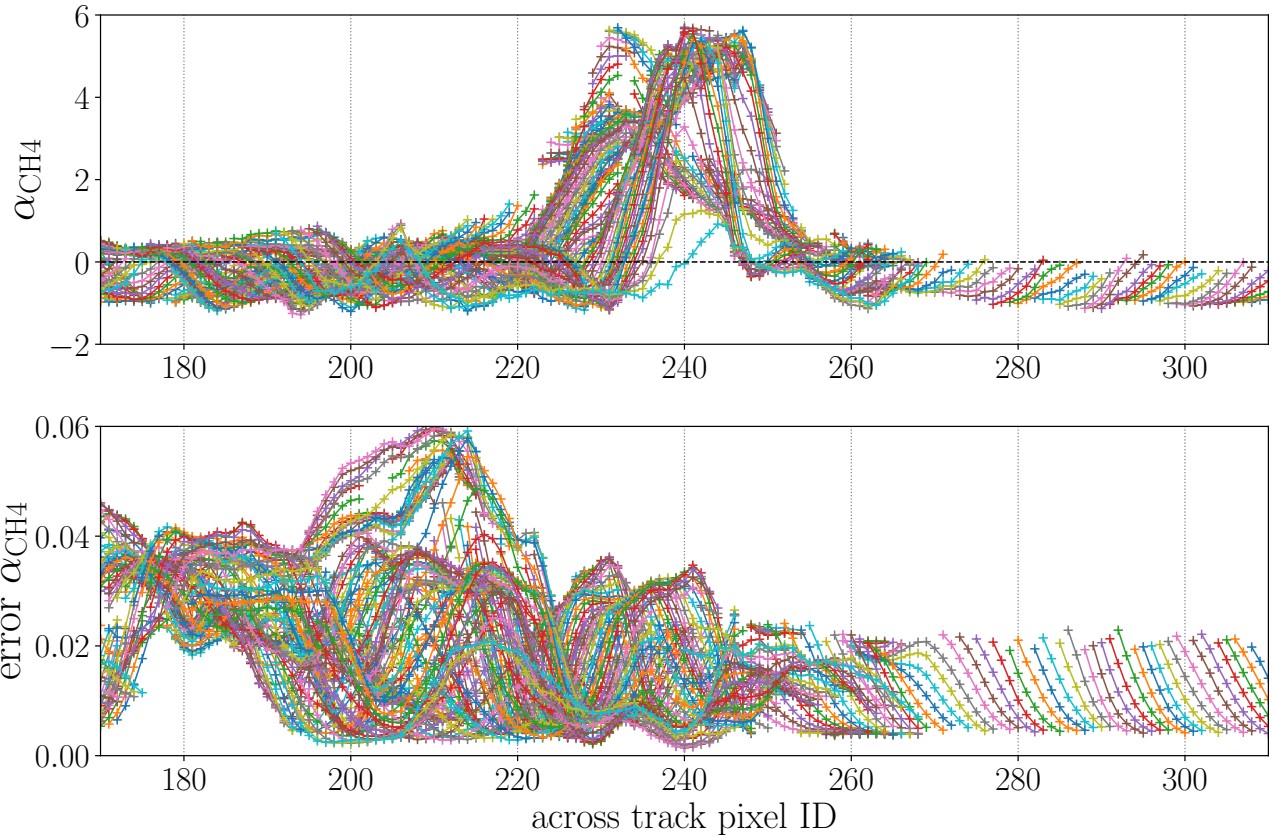

**Figure 10.** Across track depiction of the $3 \times 3$ spatially averaged multi-interval fits from Fig. 9 for the along tracks pixels 6550–6650 are depicted. The top panel shows the corresponding $\alpha_{CH_4}$ scaling factors while the errors are depicted in the lower panel. Note that filtering was applied for this figure.

The transverse section of the plume from Fig. 9 between along track pixels 6550–6650 is depicted in Fig. 10. This time the retrieval output was postprocessed so that outliers in the residual norms distribution and reflectivity coefficients $r_0 > 1$ were filtered out. The output clearly identifies stable background $CH_4$ concentrations and a significant enhancement between across track pixels 220–260 (two peaks). Variations in the plume's shape further downwind from the source can also be studied. Note that a almost twofold increase methane's total column corresponds to a $\alpha_{CH4} \approx 6$ (also see Sec. 3).





### 3.1.3 Separable least squares


The fit of single HySpex measurements with the separable least squares method turned out to be challenging as many retrievals did not converge due to the rank deficient linear problem. This confirms the findings from our simulations (not shown), which also indicate that the SLS algorithm is more sensitive to the quality of the spectra (SNR). In order to enhance the SNR of the measurements $5 \times 5$ averages were used for the separable fits. The results for the SLS fit (VarPro) for the combined 4K and 6K

intervals are depicted in Fig. 11. The retrieval output of the single window 4K and 6K retrieval mimic those of the NLS fit in Fig. 8.

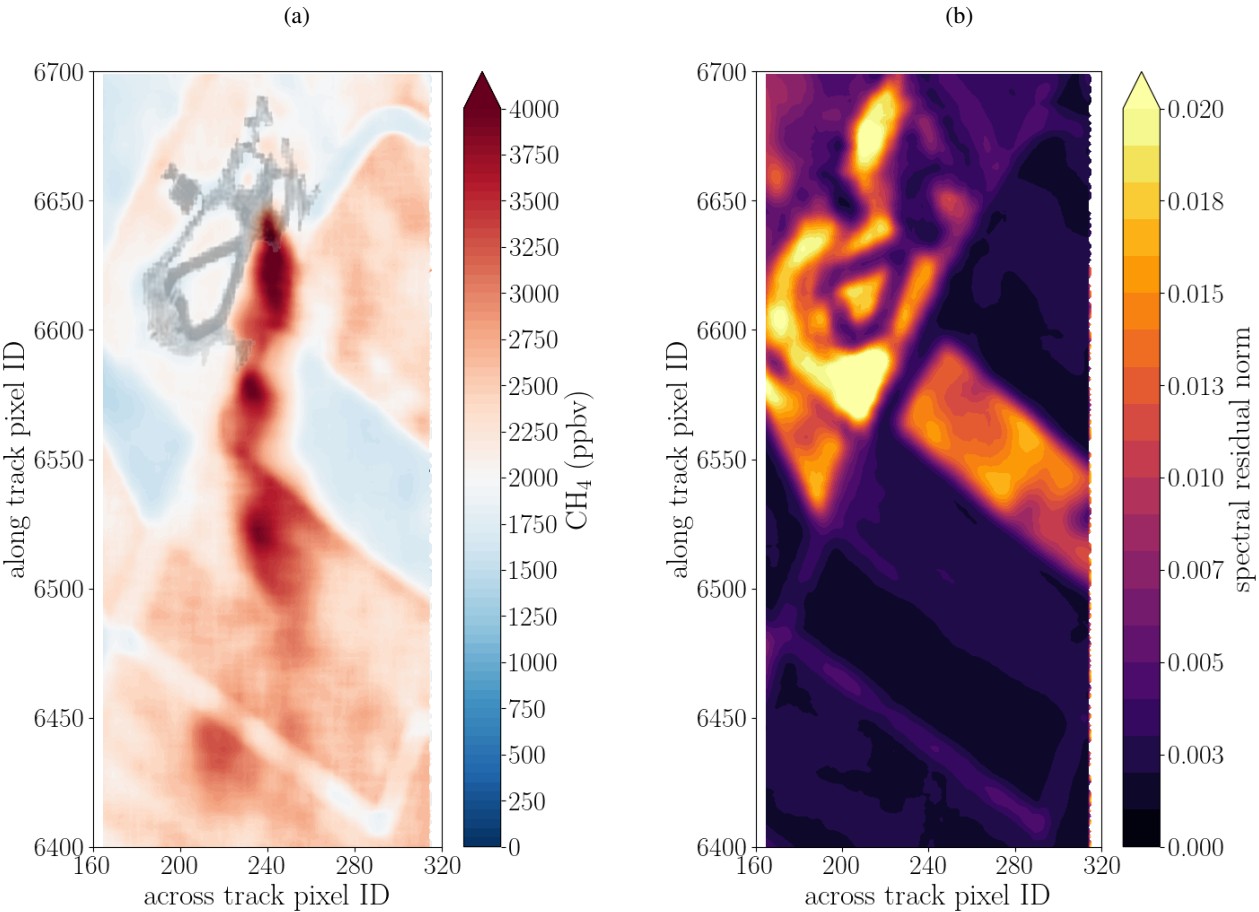

**Figure 11.** SLS fit for the combined 4K and 6K intervals for $5 \times 5$ spatially averaged observations. **(a)** Methane plume enhancement and **(b)** corresponding spectral residual norms.



### 3.1.4 Generalized least squares

In Fig. 12 the retrieved columns for the generalized least squares (GLS) fit from $3 \times 3$ averaged observations for scene 09 for the 4K and 6K intervals are shown, respectively. The algorithm employs the inverse of a scene's covariance structure to account

for background statistics in the retrieval (Thorpe et al., 2013; Nesme et al., 2020) and is well suited for detecting concentrated sources. Correlation of methane enhancement and surface reflectivity is reduced yielding a more pronounced plume signal.

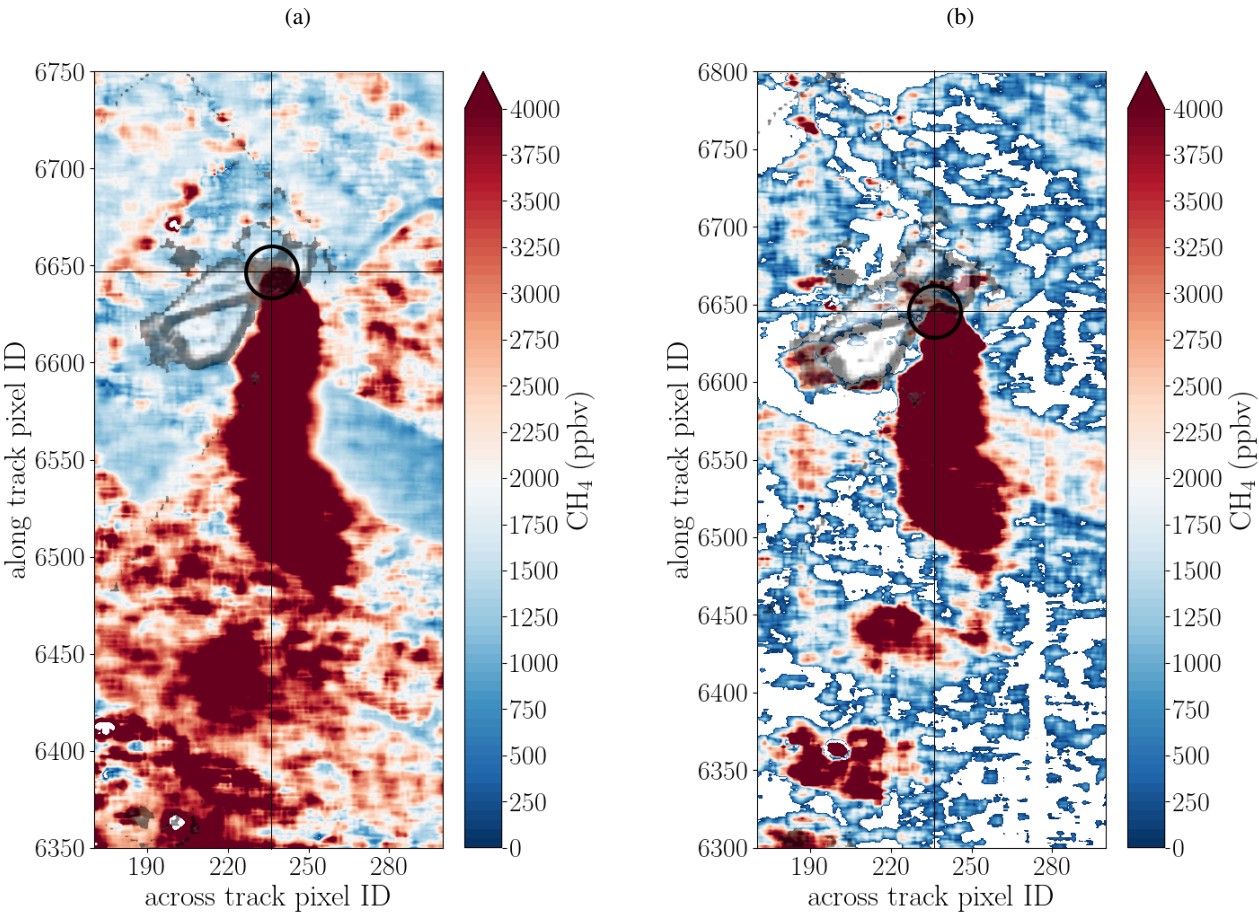

**Figure 12.** Methane plume for the GLS fit from **(a)** 4150–4900 cm$^{-1}$ and **(b)** 5700–6300 cm$^{-1}$ inferred for $3 \times 3$ spatially averaged HySpex observations. White pixels did not converge and are therefore not included in the colorbar. The fitted state vector was $\boldsymbol{x} = (\alpha_{\mathrm{CH4}}, 3r)$.

### 3.1.5 LLS

Methane concentration enhancements can be inferred with the linear retrieval scheme given the state vector and retrieval interval are properly chosen. In Fig. 13a a small retrieval interval of $\pm 50$ cm$^{-1}$ around 6000 cm$^{-1}$ with the state vector $x =$





$(r_0, b_0)$ was used. The small intervals also make the retrieval rather insensitive to variations in the ground albedo. Note that the actual enhancement factor is found by dividing the first element of $x$ by the second element, i. e. $\beta_{CH_4} = b_0/r_0$ (see Sec. 2.5). This setup was able to locate the source and also the drift of the $CH_4$ plume with the wind is traceable several hundred meters from the source. Also the enhancement factors agreed with the nonlinear fit within 20–50 % although the background concentration appears to be negatively biased when compared with the nonlinear fit. The bias turned out to be sensitive to

the width of the spectral range and increased towards larger intervals while the fit quality decreased. Moreover, the fit was only stable for small spectral intervals. More reflectivity coefficients have adverse impact on the fit as the problem becomes very ill-conditioned. Using standardized radiances as pointed out in Sec. 2.5 eliminates the need for higher order reflectivity coefficients in the linear fit and allows for larger spectral fit intervals.

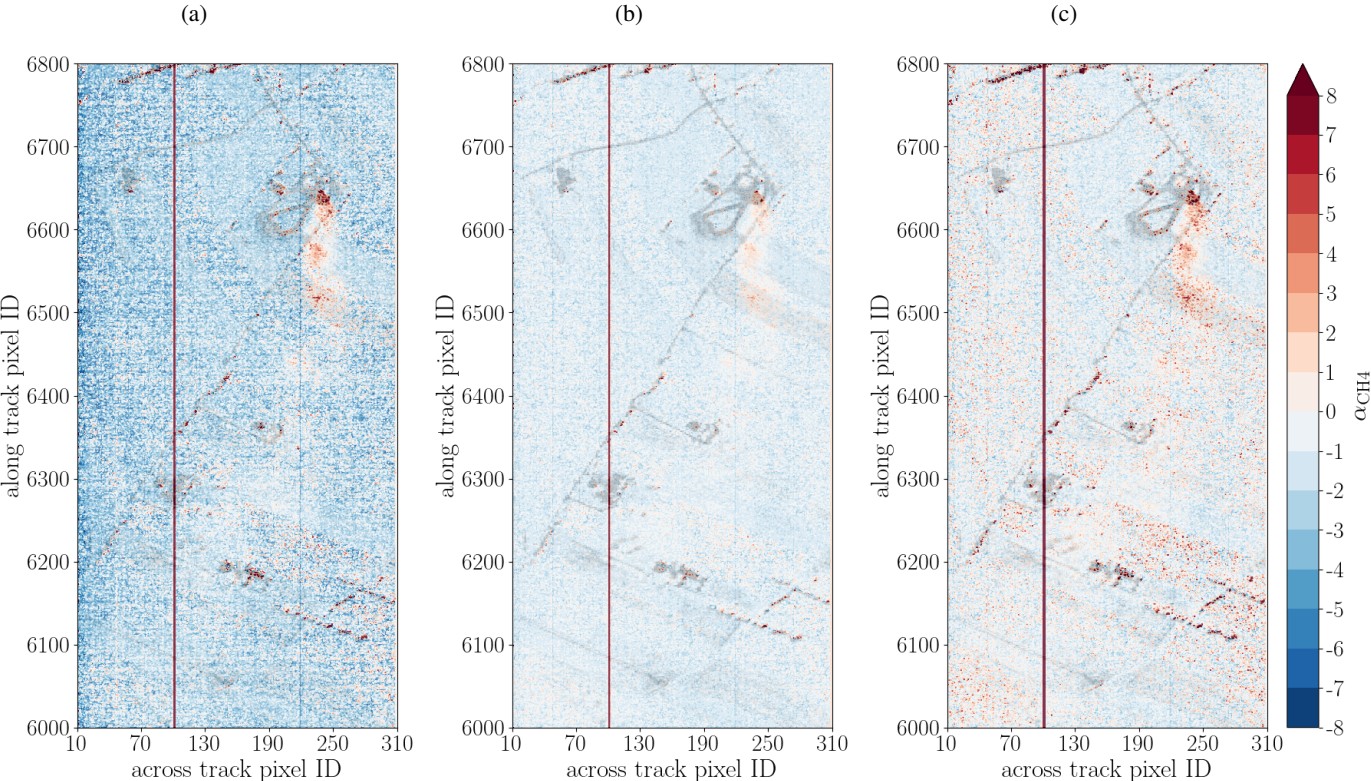

**Figure 13.** Retrieval of $CH_4$ enhancements for individual HySpex observations from scene 09. Results in the first figure **(a)** shows the linear forward model setup with $x = (r_0, b_0)$ from 5930–6080 cm$^{-1}$ while **(b)** shows the subdivision setup $x = (r_0, r1, r2; \beta_{CH4})$ from from 5900–6100 cm$^{-1}$. In the latter method the methane enhancements are less pronounced but the bias is also somewhat smaller. The NLS fit based on Eq. 6 for state vector $x = (r_0, r1, r2, \alpha_{CH4})$ is depicted in **(c)**. Note that the failed retrievals at across pixel ID 104 are caused by the pixel's bad sensitivity at 5992.74 cm$^{-1}$ (see Fig. 2 and 6). It has significant impact on the fit since the selected retrieval interval for the linear fit is small and contains few pixels (see Fig. 2).



In order to facilitate larger retrieval intervals with higher order reflectivity polynomials a slightly modified linear retrieval
setup was examined. It fits the reflectivity coefficients in the 'wings' of the retrieval window and subsequently estimates the
enhancement factor $\beta_{CH4}$ in the interval between (the center region of the retrieval window) $\boldsymbol{x} = (r_0, r1, r2; \beta_{CH4})$. The result
is shown in Fig. 13b. The setup allows to increase the spectral interval and include additional coefficients in the state vector
as the subdivision of the spectral interval avoids the attribution of variations in the $CH_4$ absorption band to the reflectivity
polynomial. However, it requires two linear least squares fits, i.e., one to estimate the reflectivity polynomial and another to
fit $\beta_{CH4}$. Note that the idea of separating pixels that belong to absorption and not was also employed in the in-band/out-band
spectral residual fits in Sec. 2.5.3.

Finally, the 'classical' NLS was applied for the same narrow spectral interval so that its result can be compared to the
linear fit. The outcome is depicted in Fig. 13c with the methane source clearly identified. Compared to the linear setup, the
nonlinear fits are more sensitive to variations in albedo but yield a smaller bias. The relative enhancement is slightly better
represented in Fig. 13a. However, in contrast to the linear fits the NLS fit is able to detect $CH_4$ enhancements for large intervals
of several hundred wavenumbers (see Sec. 3.1.2). The analysis also showed that the albedo induced variations in the NLS
are less pronounced in scene 11 which was observed at approximately twice the altitude. However, the impact of decreasing
ground pixel resolution from higher altitudes on inferred enhancements was also recognized.

## 3.2   Matched Filter

The classical and cluster tuned matched filter was examined for scene 09. Both variants clearly identifies the methane plume,
however, as shown in Fig. 14 the cluster tuning is beneficial in reducing the interference of the plume signal with surface
reflectivity.





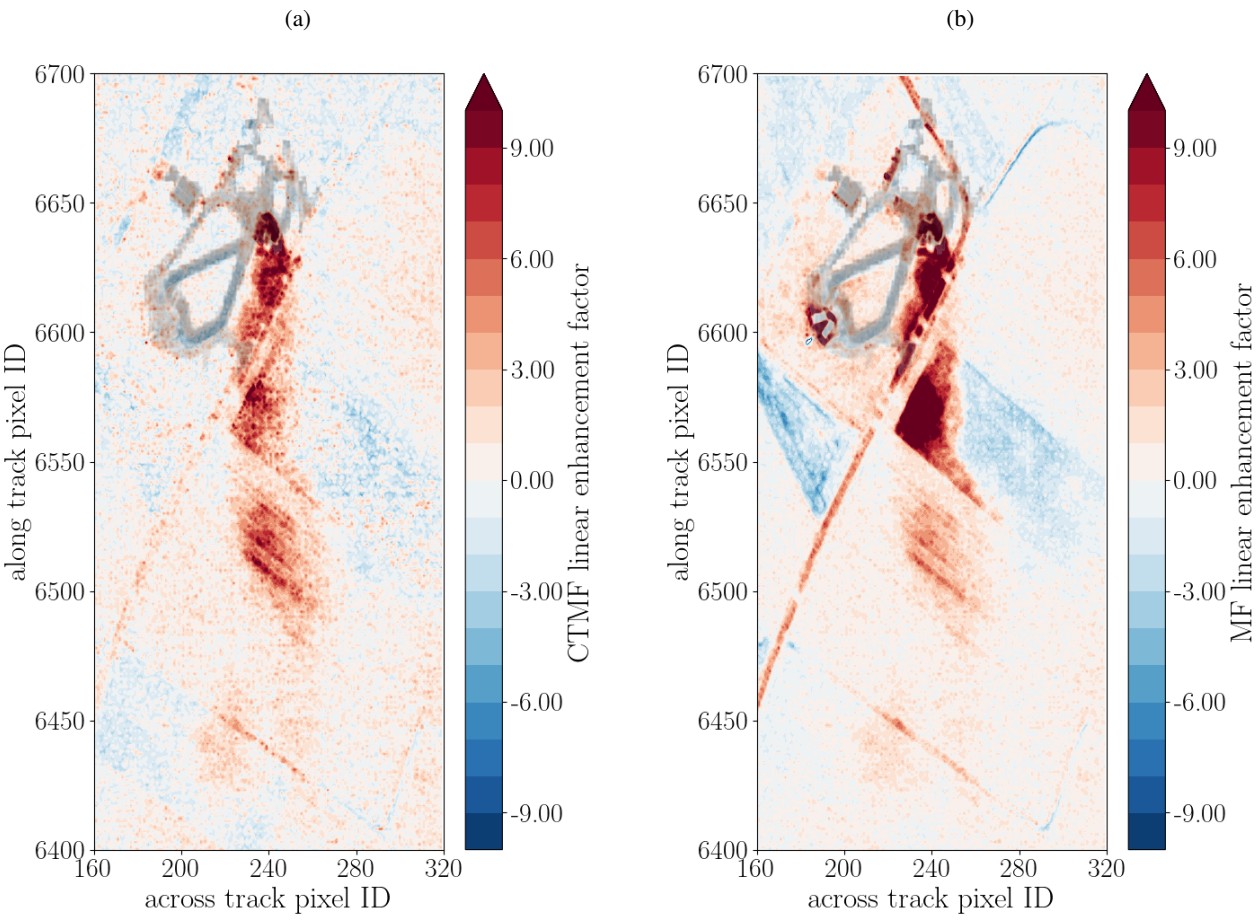

**Figure 14. (a)** Cluster tuned matched filter and **(b)** classical matched filter fits from 4150-4900 $\text{cm}^{-1}$. The enhancement factor indicates the number of mixing ratio length (390 ppm m) found in the observed spectrum by scaling its Jacobian (see Eq. 16).

## 3.3 Singular Value Decomposition

The retrieval of $CH_4$ via the SVD based method in Fig. 15 can clearly identify the methane plume. The method yields consistent
results for both spectral intervals. The retrieval setup employed the first four base vectors and the $CH_4$ Jacobian (see Fig. 7) in the linear least squares fit. Other combinations were tested but higher order base vectors were found to interfere with the methane signal so that this one turned out to give best results when using the $CH_4$ Jacobian from the model output. The plume was also identified for the purely 'data-driven' approach, i. e., where the SVD base vector that mimics the $CH_4$ absorption is used as the target signature (and does not require the forward model's Jacobian).





It was found that cluster tuning significantly improves the results when only four base vectors plus the model Jacobian are used. The reason is that variance within each cluster is smaller. Moreover, the cluster tuning is beneficial in reducing the interference of the plume signal with surface reflectivity.

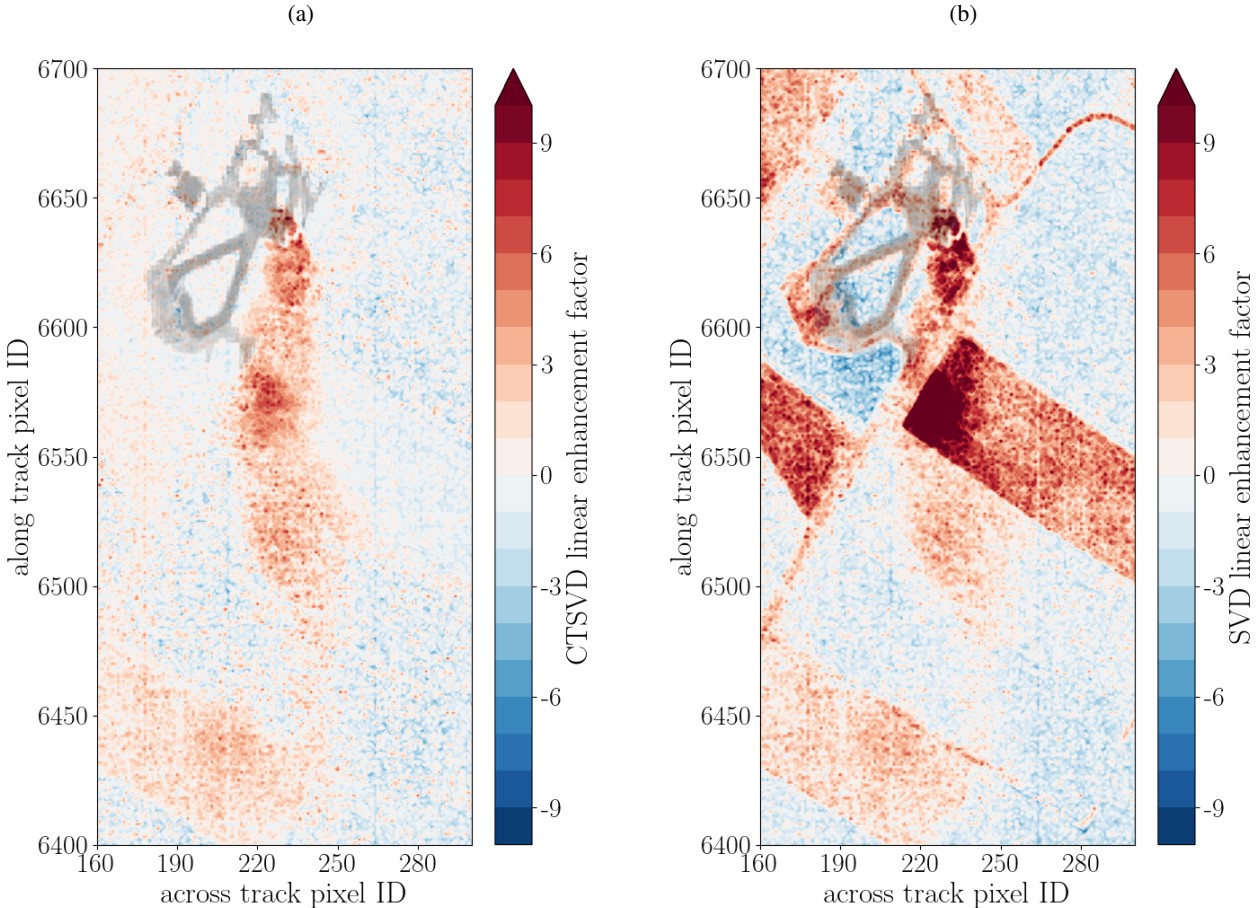

**Figure 15. (a)** Cluster tuned background SVD and **(b)** background SVD fits from 5700-6300 cm$^{-1}$. Same as in the MF fit, the enhancement factor indicates the number of reference plumes mixing ratios found in the observed signal. Methane Jacobian was calculated for lowest 2km (the plume component).

## 3.4   SSD

Finally, the SSD method for the detection of enhanced methane concentrations (see Sec. 2.5.3) is assessed. Figure 16 shows
the output from Eq. 23 for a zero (a,d,g), first (b,e,h) and second order (c,f,i) polynomial, respectively. The range of the spectral interval upon which the polynomials are fitted vary so that 5960–6040 cm$^{-1}$, 5940–6070 cm$^{-1}$, and 5920–6090 cm$^{-1}$ applies, respectively. The zero order polynomial (a constant—the mean) is rather sensitive to the chosen out-band pixels and only one





or two neighboring out-band pixels should be chosen for the polynomial fit. The higher order fits, in particular the second order, was much less sensitive to the chosen bounds of the interval and yielded enhanced in-band residuals also for larger 450 intervals. Moreover, because the out-band polynomial should capture the reflectivity and (known / not varying) absorption of the interfering molecules (i. e. H2O and $CO_2$) a polynomial up to second order was considered appropriate from the physical perspective. Fig. 16 clearly demonstrated that the second order polynomial is mandatory to capture the surface reflectivity.

(a)

(b)

(c)

(d)

**Figure 16.** The figure depicts results for scene 09 over Pniovek V in the left column and observations from scene 11 from 2.9 km altitude in the right column. The ratio of the spectral residuals for the in- and out-band pixel is depicted. In **(a)** and **(b)** residuals with respect to a mean (a model represented by a constant) from 5960–6040 $cm^{-1}$ are shown while in the second row the fit results for a quadratic polynomial from 5940–6070 $cm^{-1}$ is depicted.

The results show that all three polynomials are able to detect the enhanced $CH_4$ absorption but also indicate that the constant and second order polynomial are best in capturing features from surface reflectivity. In Fig. 16 also the zero and first order





polynomial are able to capture most of the surface reflectivity features, although the signal of the plume is relatively weak in the center plots. This result shows that the better the ground resolution (lower flight altitude) the more sensitive the residuals become to albedo variations and the higher the degree of the polynomial should be chosen.

## 4   Discussions

A validation from independent measurements is hence outside the scope of this study and should be examined in a dedicated
effort. Although measurements were taken in the Katowice area on June 7th by other instruments, none was made in the very proximity of the shafts (see Luther et al. (2022)). Nonetheless, the results from the well established MF method can be considered some sort of verification. Moreover, the SVD and MF methods were also examined with signatures from independent spectral unmixing algorithms and the results agreed well with $\approx 3\%$. As indicated in Luther et al. (2022, Fig. 4 and 6) wind was present from easterly directions which is in good alignment with the drift of the detected plume (also see Luther et al.
(2022, Fig. 4 and 6) and Fig. 1b).

    The BIRRA setup utilizing the scenes background pixel (observations not impacted by the methane plume) covariance statistics was found to be the most sensitive method for the detection of enhanced methane, although concentration within the plume is 2-3 times larger than for the classical least squares setup. So for investigating methane emissions at known locations this method is well applicable as its slow speed is not of much concern for some thousands of observations. Howerver, when
examining for potential $CH_4$ leakages on large datasets the linear solvers such as the SVD or MF are much more adequate due to their significant better speed performance. Spectral clustering of the background pixels revealed to improve the retrieval results of the linear methods by reducing the correlation of $CH_4$ with surface reflectivity. This is in good agreement with findings by Nesme et al. (2020).

    So far only narrow retrieval intervals were used for the linearized BIRRA scheme but with some foreseen modifications a
setup that allows for large and even multi interval setups is under investigation.

## 5   Conclusions

The study examines the feasibility of methane retrievals from hyperspectral imaging observations using various retrieval methods. It was found that localized $CH_4$ enhancements close to the ground can be detected and potentially quantified from HySpex airborne observations.

The BIRRA NLS fit turned out to be sensitive to spectral variations in the albedo which induced surface-type dependent (positive and negative) biases, an effect that was described by many studies using data from similar hyperspectral sensors (Borchardt et al., 2021). The albedo related correlation was also found in the single retrieval window solutions of the SLS fit, although it splits (separates) the nonlinear from linear (reflectivity) parameters. The effect was dominant for single spectral intervals but less pronounced when multiple intervals were chosen for the fit (e. g. 4K and 6K combined). The multiwindow
fits yielded retrieval errors below the maximum encountered enhancements which can be regarded significant. The GLS fit





significantly reduced the albedo bias and appears to be less influenced by the underlying surface-type. Moreover, this setup enhances the actual methane signal so that a well pronounced $CH_4$ plume is inferred. The two- to three times higher methane concentrations diminish to a one- or twofold difference when adding the surface related biases in the classical retrieval setups.

The linear estimators turned out to be very fast and hence good for near real time processing of large hyperspectral datasets.
The well established MF method for hyperspectral data agree well on the enhancement pattern and confirmed the BIRRA results. The SVD based fit confirmed the results and underlined that the identified enhancement resembles an increased signal of methane absorption. Both linear methods yielded increased performance when the scene was further divided into clusters by applying k-means in a preprocessing step. Another important finding is that both linear methods, SVD and MF, agree well on the plume's shape. It is important to note that the MF yields 50–100 % higher enhancement factors compared to the
SVD method which is attributed to the background covariance exploited in the MF method—a behavior also observed in the nonlinear fits.

The linear BIRRA setup was able to detect (and preliminary quantify) $CH_4$ enhancements, particularly in the wavenumber region around $6000\,cm^{-1}$. However, the linear results are sensitive to the selected combination of spectral interval and state vector. This is also attributed to degeneracies between the surface reflectivity and the broad band molecular absorption features.
The linearized forward model also tends to underestimate enhancements which agrees well with findings from Borchardt et al. (2021). In general, narrow retrieval intervals with only one reflectivity coefficient in the state vector turned out to constitute a stable retrieval setup in terms of detecting $CH_4$ enhancements. Nonetheless, it is a very fast retrieval scheme that can process scenes in near real time and simulations (not shown) indicated that the results improve for instruments with higher spectral resolutions so that the linear scheme should definitely be studied for measurements from other sensors.

Another simple yet effective method for detecting increased levels of methane is the SSD method. It detects relative enhancements and might serve as a real time (onboard/inflight) analysis tool for uncalibrated spectra. The detection method was able to pinpoint the source over various active shafts. Similar to the linear fit, it yielded best results for small intervals around $6000\,cm^{-1}$. As pointed out by Thompson et al. (2015) those linear methods should be considered complementary to other more complete retrieval algorithms such as BIRRA.

The sensitivity study of retrieval parameters with respect to different SNRs showed that the nonlinear and separable fits rather perform similar for different state vectors. It was shown that low SNRs in the measurement spectrum make the co-retrieval of aerosol optical depth together with a (high order) reflectivity polynomial challenging, rather impossible.

In conclusion, the presented methods are suitable to detect methane enhancements from hyperspectral SWIR observations at high spatial resolution. Moreover, the new Python version of the BIRRA code which uses Py4CAtS as its forward model
turned out to be a flexible toolbox for prototyping.

In accordance with Guanter et al. (2021) the brightness and homogeneity of the surface are major drivers for the detection and quantification of methane plumes. Also Borchardt et al. (2021) found that the retrieved total columns suffer from retrieval noise which varies significantly over different surface-types. The study also found large discrepancies in the fitted total columns of two different retrieval algorithms. It also showed that strict filtering might allow to provide enhancement values necessary
to calculate fluxes although the absolute concentrations retrieved using the different methods need to be assessed in a separate

validation study. Although greenhouse gas observations from HySpex-like sensors are challenging primarily due to its low spectral resolution further studies should investigate the potential for leakage mapping. In a next step, which is outside the scope of this study, the estimation of emission rates should be studied. Furthermore, methods specific to imaging spectrometer data such as spectral unmixing can be tested as an alternative preprocessing steps to cluster the scene for subsequent $CH_4$

retrievals based on the MF or SVD as it removes clutter while keeping unaltered the spectral information from the methane plume.

*Code availability.* The forward model is available via the Py4CAtS (Schreier et al., 2019) software suite under https://atmos.eoc.dlr.de/tools/Py4CAtS/index.html.

*Data availability.* On request

*Author contributions.* Philipp Hochstaffl (PH) developed and implemented the retrieval setups, ran all retrievals and wrote the manuscript. Franz Schreier (FS) originally designed and developed the software package Py4CAtS and supported the data evaluation. Claas Köhler (CK) conceived the experimental setup and conducted the data acquisition of the airborne measurements. Andreas Baumgartner (AB) performed the instrument calibration and Level 0-1 processing. CK, AB contributed the experimental setup to the manuscript. Daniele Cerra (DC) gave valuable advice for the cluster tuning approach and provided spectral unmixing data for the verification of the SVD and MF results. All

authors reviewed the manuscript.

*Competing interests.* The authors declare that they have no conflict of interest.

*Acknowledgements.* We thank Thomas Trautmann and Peter Haschberger for valuable criticism of the manuscript. Furthermore we thank Konstantin Gerilowski for initiating coorperation with the CoMet campaign and Andreas Fix as the campaign leader for the support and coordination.



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
