# Peer review of "Methane retrievals from airborne HySpex observations in the short-wave infrared"

_Atmospheric Measurement Techniques, 2022_

## Author Comment (AC1)

The document lists the comments (written here in small font slanted) and our responses to **Reviewer #1** .

We kindly appreciate the constructive review of our manuscript. The comments were great guidelines to improve the paper. Note that in the *Specific comments* some comments are only answered with DONE if the corrections were directly applied to the manuscript.

**General comments:**

*This article applies several different CH4 retrieval schemes to HySpex imaging spectrometer data from an anthropogenic point source. It compares a nonlinear algorithm, involving spectrum fitting using a nonlinear radiative transfer model, to various linear schemes. It also compares several algorithm variants, including pre-clustering data with k means and accounting for the covariance of surface reflectance in the nonlinear model.*

*There are some clear and obvious achievements. The algorithm descriptions are incredibly comprehensive, with enough detail to serve as a reference for investigators coding their own implementations. The manuscript deals with an important and timely topic, adding to the growing literature on point source GHG detection from coarse-spectral-resolution imaging spectrometers. It independently validates these approaches, and adds some new data sets to the mix.*

*This said, I have some recommendations for how the manuscript might be improved.*

*My first major recommendation is to clarify the thesis statement. After reading the article, I'm still a bit unclear on the fundamental contribution. The manuscript focuses mostly on retrofitting the BIERRA algorithm for CH4 point source detection at coarse spectral resolution. However, by the end of the manuscript it is unclear what advantages this offers beyond the state of practice Matched Filter methods or other very similar nonlinear model-fitting methods in common use (like the Thorpe et al. IMAP-DOAS approach). The affect of albedo on nonlinear CH4 retrievals is great, but it has been investigated even more thoroughly before - see for example Ayasse et al. 2019 (https://doi.org/10.1016/j.rse.2018.06.018). I think the authors could do a better job of calling out what is new and significant about the BIERRA approach.*

Firstly, we added a clearer statement of the research question (see before the Methodology section) to highlight the contribution of our work. Specifically, we have emphasized the retrofitting of the BIRRA algorithm for including covariance matrices in the fit which turned out to be a crucial aspect for fitting methane from hyperspectral datasets. In the revised manuscript the advantages of the weighted BIRRA solver over other methods is pointed out more clearly. Moreover, it is now also discussed in more detail how BIRRA compares to other methods (see Discussion section) and the Result section clearly highlights its superiority in terms of accuracy and precision compared to the other setups.

*My second major recommendation is to have a quantitative performance comparison. The current assessment is fairly subjective, related to the quality of the plume image and the visual appearance of background interference. Couldn't the background variability outside the plume be used to quantify the detection noise for each method? And couldn't the strength of the plume enhancements then be used to create an SNR score or statistical confidence? As a part of this effort, it would be great to translate all of the plume maps into similar units. Currently maps appear variously as ppbv, alpha-CH4, and "enhancement factors," which makes it difficult to inter-compare. It should be possible to translate any one of the linear detection algorithm results into an equivalent CH4 mass enhancement, and compare the effective plume-to-background detection SNR of each of the algorithms.*

To fulfill the request, we performed a more quantitative performance comparison by utilizing the background variability outside the plume to quantify the detection noise for each method. As part of this effort, we provide the same units for the CH4 enhancements (mole fractions) throught the results section, which makes it easier to compare the performance of different algorithms. Furthermore, we have created a score to evaluate the quality of the solvers (see Tables 2 and 3).

**Minor suggestions**:

*1. Almost all of the prior literature cited on CH4 point source detection, and the vast majority of the imaging spectroscopy community working at these spectral resolutions, plot spectra in wavelength rather than wavenumber. Setting aside the question of which convention is more convenient or appropriate from a technical perspective, it would certainly be easier for the majority of the readership to quickly understand the figures if wavelengths were used. This would make the instrument sampling evenly-spaced in the horizontal direction.*

We revised the corresponding figures to present the spectra in wavelength units in addition to

wavenumber units. Wavenumbers are still around in the revised verison, e. g., for the naming of the retrieval windows but we think this does not compromise comprehension.

*2. On line 61, are there citations for CarbonMapper or CO2Image missions? I think the claim that CarbonMapper operates at higher spectral resolution than average land surface imaging spectrometers (5-10nm sampling) could be incorrect.*

Since we did not find reliable information on the spectral resolution of CarbonMapper we removed the instrument from the introduction.

*3. On line 65, in the literature review of airborne CH4 point source campaigns, consider also the studies by Frankenberg et al. 2016 (https://doi.org/10.1073/pnas.1605617113) and Duren et al. 2019 (https://doi.org/10.1038/s41586-019-1720-3) which were earlier and larger.*

Both source were added in the literature review of airborne CH4 point source campaigns.

*4. Figure 2 (a) seems to be missing some lines. "Grass" is misspelled.*

DONE

*5. On line 183, can you provide any more specifics about the low order polynomial used? What was its degree and where was it centered? The details are significant because, as you note, the surface reflectance is often quite complex over these wide spectral ranges. Changes in the reflectance representation can have huge changes on albedo sensitivity.*

The last paragraph in Sec. 2.3.2 answers this question (second order).

*6. On line 195, the section comparing different least squares solvers seems ancillary. Least squares solvers are a commodity*

DONE

*7. Figure 10. I'm not sure what this is supposed to show. Could it be removed?*

DONE

*8. On Figure 12, the enhancement within the plume appears completely saturated, which makes it difficult to assess. Can you rescale the colormap to make it more similar to the other plume images?* We adapted and rescaled the colormaps. They are now adjusted to a common scale to facilitate comparability. Most pixels should be within the colorbar range.

*9. The discussion and conclusion is good, but would be further strengthened by quantitative claims about where and how the different algorithms outperform each other.*

We modified and enhanced the discussion and conclusion sections. We added Tables 2 and 3 and a correlation plot to clearly present results more quantitatively and highlight areas where each algorithm outperforms the others.

With best regards,

Philipp Hochstaffl and co-authors

---

## Author Comment (AC2)

The document lists the comments (written here in small font slanted) and our responses to **Reviewer #2** .

We kindly appreciate the extensive review of our manuscript. The comments were great guidelines to improve the paper. Note that in the *Specific comments* some comments are only answered with DONE if the corrections were directly applied to the manuscript but all the aspects were addressed if not otherwise stated in an answer.

**General comments:**

*The manuscript of Hochstaffl et al. present an intercomparison of different CH4 retrieval approaches for airborne hyperspectral measurements during an overflight of the HySpex instrument over a known CH4 emission source in Poland. Gives a short overview over the different techniques and present the retrieved maps for specific scene which are then discussed. CH4 retrievals from hyperspectral measurements is a quickly evolving field and there is merit in intercomparing different retrieval methods based on real, measured spectra.*

*The manuscript is in principle relevant and suitable for Atmos. Meas. Tech. but there are several shortcoming that limit its value. The manuscript is somewhat dominated by the method section while the discussion could be more detailed. In this section, a lot of material is squeezed in which makes it hard to read and I am not sure how much the reader gets out of it, especially since variables are not always well defined or used in consistent manner. The presented analysis is purely qualitative, very brief and limited to a single scene. The lack of ground truthing means that it is not possible to tell which method is best and thus the study is limited to comparisons against each other which then should be done more rigorously.*

The revised manuscript has successfully addressed the issues that were identified in the initial feedback. The authors have made an effort to balance the method section with a more concise and well-structured discussion and conclusion. The material has been rearranged in a way that hopefully makes it easier to read and understand, and the variables are now well-defined and consistently used throughout the manuscript. The analysis has been expanded to include multiple scenes (scene 09 and scene 11). The authors has taken into account the feedback fo do a comparison of the method against each other more rigorously by adding two additional sections to the manuscript: Sec. 3.3 titled *Statistical significance of results* and Sec. 3.4 *Errors and correlations*. Both analyze the retrieval results in a more rigorous manner.

*Main comments are:*
*Several figures need to be improved and labelled consistently (see detailed comments below). Especially the map of retrieved methane should be shown for the same number of cross and along track pixels so that results can be compared. Also, give results in CH4 for all methods and not as scalding factors.*

The figures were improved so that they are labelled consistently, making it easier for readers to understand and compare the results. In particular, the maps of retrieved methane are now shown for the same number of cross- and along-track pixels, allowing for a easier comparisons. Moreover, results are now consistently presented in terms of CH4 mole fractions for all methods, rather than as scaling factors etc. The improvements should definitely enhance the readability of the study.

*The analysis should be more quantitative. In the discussion, you state that the MF method can be used as reference, so lets also do this. You can apply simple methods to identify a plume (eg thresholding) which will then allow you do contrast inferred plume shapes. Also, I would like to see correlation plots of CH4 enhancements between different methods for example for pixels within the plume (eg defined according to the the MF method).*

The revised manuscript has successfully addressed the issue of the lack of quantitative analysis. The authors has applied the t-test method (Sec. 3.3) on the rerieval results from different methods to identify the plume and compute statistics upon. Additionally, the author has provided correlation plots of CH4 results between different methods. These changes should allow readers to better understand the relationships between the different methods.

In course of the revision of the manuscript we decided to not use the MF method as a benchmark but do initially not prefer a method and simply compare each against all others. It was found

that the nonlinear covariance weighted BIRRA retrieval yields best results in terms of contrast and with the smallest statistical uncertainties.

*Also, check all equations and ensure that all variables are fully defined and used consistently. For example, beta is used to describe 3 different variables.*

In the revised manuscript the issue of inconsistent variable is resolved. All variables are fully defined and used consistently throughout the manuscript. Hence it should be easier now to follow the methodology and the subsequent analysis. If the authors overlooked a case we kindly ask to report it.

**Specific comments:**

*Page 2, line25: important greenhouse gas -¿ important anthropogenic greenhouse gas*

DONE

*Page 2 line 32: fossile -¿ fossil*

DONE

*Page 2 line 38: besides satellites, there are the global in-situ surface networks*

The networks Global Atmosphere Watch (GAW) and Integrated Carbon Observation System (ICOS) were added

*Page 2 , line 55: smaller emitting area. This does not refer to IR so it is reflecting rather than emitting.*

Corrected this to . . . loss of photons caused by the smaller ground pixels . . .

*Page 3, line 61: add reference for CHIME, eg. Rast et al., IEEE, 2021*

DONE

*Page 3, lines 60-63: add MethaneSat for completeness*

DONE

*Page 3, line 75: slowly varying part is also from scattering*

DONE

*Page 3, line 81: This study compares various retrieval schemes . . . -¿ please add the goal of study*

We added a statement before the methodologs section.

*Page 4, section 2.1: I suggest to add a table with the key instrument parameters for the HySpex instrument*

DONE

*Page 4, line 96: in detail in (IMF) -¿ missing reference ?*

Corrected

*Page 4, figure 1b: The fonts on the figure are too small and can not be read in a hardcopy.*

Adapted the figure.

*Page 4, line 97: in the following chapters -¿ in the following sections*

Not applicable anymore.

*Page 4, line 101: . . . and if so how accurate . . . -¿ I don't think that this is addressed in this manuscript ?*

Reformulated since an accuracy assessment would required some sort of truth reference.

*Page 4, line 108: seen in Image 1a, -¿ Figure 1a*

DONE

*Page 5, line 109: wind data for the USCB area -¿ This needs to be defined separately to the definition in the abstract. However, I don t see the need to introduce this acronym as it is not used anywhere else.*

It is used three times and we decided to keep the acronym but introduce it only in the text not in the abstract.

*Page 5, line 119: 967-2496 nm (4005-10338cm-1). Throughout the manuscript at some places wavelength and and at other wavenumber is used. Since wavenumber is primarily used, I suggest to use consistently wavenumber throughout the manuscript and give everything in wavenumber and not wavelength.*

In accordance with comments by other reviewers, it was decided to introduce wavelength but keep wavenumbers in particular for the naming of the retrieval windows since e. g. absorption lines in molecular spectroscopy databases are and hence cross sections are usually given on wavenumber grids.

*Page 6, figure 2: a) please make fonts larger and lines thicker. In a hardcopy this figure is hard to read. Also, can you give a reference for the albedo data*

This figure was compiled from another person and will be updated in the final version before possible publication.

*b) Use either 'wavenumber' or 'wavenumbers' for the x-axis. Use either round or square brackets to give units. Label the y-axis with the shown parameter (not only units).*

*Remove the 2 extra digits on the x-axis labels. Caption: CH4 -¿ CH4*

*I don't think that adding the vertical lines to indicate spectral pixels adds value and neither does overplotting all spectra into the figure. I suggest to remove the lines and show a mean spectrum and a standard deviation. You could add the one spectrum with the outlier.*

The figure was adapted according to the reviewers suggestions.

*Page 6, line 131, see absorption from methane's 2v3 band around 6000cm-1 -¿ where do I see this. Can you label this in the figure ?*

This statement was removed for the sake of clarity and since it is not clear whether the feature represents the methane's 2v3 band absorption.

*Page 7, line 136: under clear sky conditions (cloud free) -¿ also scattering free in general*

DONE

*Page 7, eq. 1: ds needs to be removed in the sum of the first equation. define all variables including tau, nu, p, T, s and m*

DONE

*Page 7, lines 145-150: I don't see how the extract on aerosol optical properties is relevant. I suggest to remove this and simply refer to a textbook. The use of wavenumber of wavelength in this section is unnecessary. K_aer: give units.*

The paragraph was removed and a reference with the formulas therein added.

*Page 7, line 153: composed by pure -¿ composed of pure*

Not applicable anymore.

*Page7 , lines 154-155: define z, also I don't see the need to use zmol, zsc instead of simply z.*

DONE

*Page 7, eq.5, define tau_bg and tau_pl. What is alpha here ?*

Variable tau_bg is now explained and tau_pl changed to tau.

*Page 8, line 161: The CH4 background as well as the CO2 initial guesses -¿ The CH4 background profile as well as the CO2 background profile*

DONE

*Page 8, figure 3: BoA, TOA -¿ BOA, TOA; Al least for CO2 and CH4, a mixing ratio profile would be more meaningful*

Not applicable anymore.

*Page 9,line 186: is SRF different to ISRF ? This are examples for the many acronyms introduce but not used in the manuscript.*

The use now ISRF throughout the manuscript.

*Page 10, figure 4: remove unnecessary digits on x label. Thicker lines in panel b would be helpful*

DONE

*Page 10, line 194: Jacobian matrix -¿ define Jacobian*

DONE

*Page 11, Figure 5: give the definition of alpha and r in caption . Remove unnecessary digits*

This plot was removed for brevity and as other authors deemed it not necessary.

*Page 11, line 202: the converged spectrum-¿ the converged spectrum I*

DONE

*Page 11, line 206: from the diagonal elements -¿ from the square root of the diagonal elements*

DONE

*Page 12: beta is already used as Angstrom coefficient on page 7. Please use another variable name here.*

DONE

*Page 12, eq. 11: define meaning of y hat.*

The variable was substituted by $\vec{y}$ for consistency with the description of other methods.

*Page 13, define alpha tilde is.*

The variable is explained. However, note that in course of the revision it was decided to formulate the light path correction according to literature mentioned in Sec. 2.3.2 and disregard the difference in methane and carbon dioxide in the lowest layer.

*Page 13, line 231: What is a scene average scaling factor. I don't believe that the given references apply such a scene average scaling factor.*

The aim was to provide references that also account for scattering by the proxy method. In our setup the co-retrieval of CO2 with methane was problematic hence this approach was chosen.

*Page 13, eq. 12: usually the CH4 to Co2 ratio is multplied by a 'known' CO2 profile. If you use the co2 scaling factor directly as a correction for light path modificatiins then you assume that the CO2 profile is perfect and that alpha is 1 in absence of scattering.*

This is true and the assumption is now explicitly stated in the manuscript.

*Page 14, line 256: the (saturated, see . . .-¿ the saturated (see . . .*

DONE

*Page 14: eq 14: what is the meaning of the up and down arrows in the optical depth. Why is there a beta factor in the*

*Taylor expansion why is not in the exp function. Also, how is beta defined. Note that beta is used already twice with other meanings.*

Arrows are representing the up- and downward paths but the notation was changed to simply $\tau$. Also $\beta$ is not used anymore in this context.

*Page 14, lines 261-261: M and N is now used in capital but was used before in small letters m and n (page 11)*

Capital $M$ now only represents measurements and $m$ is a variable for a certain species (molecule).

*Page 14, line 269, condition number of 885 -¿ Please put this in some context. Which condition number is sufficient and which not.*

The discussion on condition numbers was removed as other reviewers question if this part is necessary. Moreover the nonlinear fits are now made with the simple state vector $\vec{x} = (\alpha, r_0, r_1, r_2)$.

*Page 15, eq. 16: isn't (J - mu) the target spectrum t . If so, then use t in equation.*

DONE

*P15, l284: here tau and beta is defined which would already be needed with eq. 14*

This has been resolved but some quantities such as e. g., $\alpha$ are specific to the used methodology, however, it should always represents the enhancement factor.

*P 15, eq. 18: is t here now the Jacobian. Jacibas so far is called J while t has been used as target spectrum in section 2.5.1.*

The variable J is now only used for the Jacobian matrix in the nonlinear fit and t is used for the target signature in the MF and SVD schemes.

*Page 16, figure 7: What does stand. rad. mean? State in caption that u1-u4 are singular vectors ?*

DONE

*Page 17, eq. 21: meaning of beta ? I assume this is again different to the 3 previously define beta's?*

The variable $\beta$ now only represents the aerosol exponent.

*Page 17, line 343: retrieal's -¿ retrieval's*

DONE

*Page 18, line 351: The state vector x = (3m,3r) was found to be robust toward low SNR . . .-¿ what do you mean by robust?*

Findings from the analysis with simple simulated measurements were found to be not applicable to the actual HySpex observations for the majority of cases so that this analysis was removed from the manuscript.

*Page 18: lines 348: changing the resolution will typically also change the SNR*

For the sake of brevity and because only a simple state vector $\vec{x} = (\alpha, r_0, r_1, r_2)$ was finally used in the actual retrievals, the section on the feasibility of state vecotrs was removed.

*Page 18, table 1: condition numbers need to be put into context.*

This analysis of condition numbers for different state vectors was deleted for the above mentioned reason.

*Page 18, lines 354: use wavenumber here and not wavelength so that it is consistent*

DONE

*Page 20: figure 8: can you add figure with retrieved surface reflectivity.*

Since the surface reflectivity coefficients are regarded an effective parameter which captures all smooth components it was decided to not devote them separate plots. In the error plots the spectral residual norm is a proxy for the albedo which is usually larger over bright areas.

*Page 21, Figure 9 : make figure the same along track and across track range as figure 8. Please make all the plume figures the same range for comparability. Also, give CH4 on maps and not the scaling factors (figure 13 and 14, 15).*

DONE

*Page 22, figure 10: Hard to see anything. I suggest to plot only lines instead of lines+symbols.*

Not applicable anymore. Plot was removed as the 2D-maps contain this information and provide a more complete picture.

*Page 26, lines 424: The relative enhancement is slightly better represented in Fig. 14a.-¿ I don't think you can tell what is better as you don't know the truth.*

A more rigorous statistical assessment for plume identification and correlations between different retrieval methods is now performed.

*Page 27: The method yields consistent results for both spectral intervals. -¿ results shown in Figure 16 are very different. In which way are they consistent ?*

Consistent with respect to the background since retrieval methods are expected to yield values close to the initial guess for background pixels (not impacted by the plume), rather independent of the spectral interval used.

*Page 28, line 444, should Figure 16 have 3 rows for zero, 1 and second order ?*

The analysis was reduced to zero and second order polynomials in the 6K window.

*Page 29, Figure 16: can you use same colour scale and format to increase the comparability ?*

DONE

*Page 30: A validation from independent measurements is hence outside the scope of this study -¿ A validation from independent measurements is outside the scope of this study*

DONE

*Page 30, lines 461: the results from the well established MF method can be considered some sort of verification -¿ what is the justification for this. Is there a reference that can be used in support ? Also, the results from MF are not used as reference for the analysis.*

In course of the revision of the manuscript we decided to not use the MF method as a benchmark but do initially not prefer a method.

*Page 30, line 463: and the results agreed well with $\approx 3\%$. -¿ where is this shown ?*

The spectral unmixing is now only briefly described as a purely data-driven approach in the SVD section. However, it is not shown and put into perspective in the actual analysis of the manuscript. Moreover, it would require another method to be introduced and assessed.

*Page 30, line 467: was found to be the most sensitive method for the detection of enhanced methane -¿ how is this conclusion drawn ?*

This statement is now underpinned by the statistical analysis in Sec.3.3.

*Page 30,line 478: and potentially quantified from HySpex -¿ you have not shown this.*

With respect to errors which is now consolidated in Sec. 3.3 and 3.4.

*Page 31: line 494: agree well on the plume's shape. -¿ this is not shown in the manuscript*

A more quantitative analysis of the plume's shape is now inferred from a t-test (see Sec. 3.3).

*Page 30, conclusion: can you discuss if the findings from this study are only applicable to Hyspex or also to wider range of hyperspectral sensors for example on satellites.*

DONE

With best regards,

Philipp Hochstaffl and co-authors

---

## Author Comment (AC3)

The document lists the comments (written here in small font slanted) and our responses to **Reviewer #3** .

We kindly appreciate the review of our manuscript and the comments were great guidelines to improve the paper. Note that in the *Specific comments* some comments are only answered with DONE if the corrections were directly applied to the manuscript but all the aspects were addressed if not otherwise stated in an answer.

**General comments:**

*This work study the performance of methane retrievals deduced by non-linear and linear methodologies from data obtained by airborne HySpex observations. Methods are applied in several spectral ranges in the SWIR, where methane absorption features are located.*

*Within non-linear methods we find the Nonlinear Least Squares, the Separable Least Squares and the Generalized Least Squares and within the linear methods we find the Linear Least Squares, the Matched Filter, the Single Value Decomposition, and the*

*Spectral Signature Detection. While non-linear methods are more time-consuming and get a best estimate, linear methods are faster and can be more suitable for real-time onboard measurements.*

*This study is helpful in order to understand the limitations of HySpex in detecting methane emissions with several methods. A good understanding of these limitations can establish a strategy to get optimal methane concentration maps on real time and after the flight.*

*The results introduced in this work are of remarkable interest and a great amount of work must have been involved. Methane retrieval and methane retrieval error figures are very self-explanatory and visual. Moreover, there are a great diversity of methodologies that have been explored, which is a decision that helps to determine more thoroughly the limitations of Hyspex for methane mapping.*

*I find high value in the objectives of this paper and the figures, but I see strong shortcomings that make me decide to accept this paper with 'major revisions'.*

*Major revisions:*

*The paper is hard to read: there are very long sentences that are difficult to understand, and I find a strong lack of coherence and consistency in writing. I would recommend a rewriting that makes the work easier to read.*

In a great efforts the issues related to the readability of the paper were addressed. Many parts of the manuscript were rewritten to improve coherence and consistency, hopefully resulting in a more understandable text.

*Figures are not exploited. Although the figure can be mentioned in the text, there is little feedback between text and figures. This happens with Fig. 3, Fig. 4, Fig. 6, Fig. 7. I doubt if these figures are necessary. Besides, I find information that does not contribute to the paper, such as a lot of details about the in the aircraft measurements, determining the nature of absorption features (vibrational transition), etc. Altogether, the paper could be shorter and preserve the important information at the same time.*

In the revised version we adds more relevant information in the captions and try to reference the figures more clearly in the text. After review of the figures led to the decision that Fig. 3 and 5 can be removed without compromising comprehension while all other figures in the manuscript were revised according to the reviewers suggestions and to hopefully better align them with the text. The elimination of irrelevant information should also contribute to a more more focused and streamlined paper.

*- Methods are not clearly introduced. Some parameters are not explained and some formulas appear without a previous justification or citation. As a consequence, the reader could doubt about the theoretical basis of the diverse methods. There is also a lack of consistency in nomenclature: some variables are written in different ways along the study, which makes it difficult to keep up with the paper.*

To improve this aspect, we carefully reviewed the manuscript and made necessary adjustments to ensure that variables are written in the same way throughout the study. This was also a big concern of Reviewer #2. We hope that the revised version now uses a consistent nomenclature, and that sufficient references are provided to support the theoretical basis of the diverse methods.

*Results are nor exposed nor discussed in a consistent manner. For example, methane retrievals are showed in both single spectral intervals and also the multi-interval in the NLS, but the multi-interval is not shown in GLS. Besides, I think a more thorough discussion would have been appropriate. A table gathering statistic information about the performance of every method could be an adequate manner to do it.*

We made sure to show the methane retrievals in a consistent manner across all methods, including both single spectral intervals and the multi-interval approach in both NLS and GLS. We have also added a table summarizing the performance statistics of each method to provide a clearer comparison of their performance.

Additionally, we have revised the discussion section to provide a more thorough and consistent analysis of the results, including correlations between methods and a more detailed discussion of the strengths and weaknesses of each approach.

*Conclusions about the different methods are not clear. Which are then the best methods? Which is the best strategy to map methane in real-time flight and after the flight? Maybe this could be clearer with the table that I commented previously.*

A table for the nonlinear and the linear schemes that summarize the performance of each method is now provided. Moreover a statistical analysis is performed in Sec. 3.3. This should make it easier to compare the methods and determine the best methods for mapping methane in real-time flight (performance is key) and post-flight (accuracy and precision is key). The strengths and limitations of each method are now hopefully better highlighted.

**Minor revisions:**

*Line 33: Fossil fuel exploitation is responsible for 30-42% of all anthropogenic CH4 emissions (Saunoise, 2020).*

I am afraid but I was not able to find this reference. Could you please provide the DOI so we can include it.

*Line 40: the absorption spectral ranges are not correct.*

Reformulated to: . . . around $1.6\,\mu$m and and $2.3\,\mu$m.

*Line 54-57: example of too long sentence.*

Splited sentence.

*Line 75: moderate spectral resolution is defined as '¿1nm'. And what about the coarse spectral resolution?*

Do you mean the coarse spatial resolution in addition to the coarse spectral resolution? The aspect is described in Line 51.

*Line 77: I think you can make a more thorough distinction between data-driven methods and physically based methods (see Guanter, 2021).*

The updated Methodology section now better describes the differences between the methods.

*Line 96: 'a VNIR-1600 and a SWIR-320m-e'. I supposed the former can measure VNIR radiation and the latter SWIR radiation, but this is not explicitly explained.*

This is described in the reference provided.

*Line 112: I think '0955 UTC' is not a valid timestamp format.*

Changed to '09:55 UTC' but without seconds.

*Line 156: 'methane enhancement' instead of 'Gaussian plume'. The gaussian plume would be the result of the methane enhanced pixels close to an emitting source.*

*Line 166: 'BIRRA level 2 processor' could be in italics.*

DONE

*Line 166: DLR initials are already explained.*

DONE

*Line 319: interpolated band ratio (CIBR) from Green et al.(1989)...*

DONE

*Line 344: retrieval.*

DONE

*Line 399-400: 'The algorithm employs the inverse of a scene's covariance structure to account for backgorund statistics in the retrieval'. This was already stated in 'Methodology'.*

Removed the reoccurrence.

*Line 445: What is (a,d,g), (b,e,h), and (c,f,i)? It is not clear.*

Not applicable anymore.

*Line 469: However.*

DONE

*Line 474: So far, only narrow...*

Not applicable anymore.

With best regards,

Philipp Hochstaffl and co-authors

---

## Author Response (AR1)

DLR — Remote Sensing Technology Institute
82234 Oberpfaffenhofen
Germany

April 2023

Dear AMT Editorial Board and Reviewers,

first of all we would like to thank the referees for the constructive and friendly review. Together with this reply we submit the revised manuscript entitled *Methane retrieval from airborne HySpex observations in the short-wave infrared* for publication in the EGU Journal AMT. The constructive criticism and useful comments have been instrumental in enhancing the manuscript's overall clarity. The manuscript has undergone substantial revisions in response to the reviewers comments received during the discussion phase. More specifically, the manuscript has been consolidated in some areas and expanded in others, as suggested by the reviewers' comments. Two new sections were created, i. e., Sec. 3.3 titled *Statistical significance of results* and Sec. 3.4 *Errors and correlations*. Both analyze the retrieval results in a more rigorous manner as requested by the reviewers. More details on changes made in response to the reviews are described in the individual response letters.

With best regards,

Philipp Hochstaffl and co-authors

---

## Author Response (AR2)

The document lists the comments (written in small font) and our responses to **Reviewer #3**.

**General comments:**

*This work study the performance of methane retrievals deduced by non-linear and linear methodologies from data obtained by airborne HySpex observations. Methods are applied in several spectral ranges in the SWIR, where methane absorption features are located. Within non-linear methods we find the Nonlinear Least Squares, the Separable Least Squares and the Generalized Least Squares and within the linear methods we find the Linear Least Squares, the Matched Filter, the Single Value Decomposition, and the Spectral Signature Detection. While non-linear methods are more time-consuming and get a best estimate, linear methods are faster and can be more suitable for real-time onboard measurements. This study is helpful in order to understand the limitations of HySpex in detecting methane emissions with several methods. A good understanding of these limitations can establish a strategy to get optimal methane concentration maps on real time and after the flight. The results introduced in this work are of remarkable interest and a great amount of work must have been involved. Methane retrieval and methane retrieval error figures are very self-explanatory and visual. Moreover, there are a great diversity of methodologies that have been explored, which is a decision that helps to determine more thoroughly the limitations of Hyspex for methane mapping. I find high value in the objectives of this paper and the figures, but I see strong shortcomings that make me decide to accept this paper with 'major revisions'.*

Thank you for your thoughtful critique and feedback on our manuscript.

**Major suggestions:**

*I find that the manuscript is really hard to read because several reasons:*

- *It is long. Shortening the manuscript could make the text more accesible for readers. I would suggest the authors to optimize the content to the truly important data.*
- *There are a lot of information that could be removed without affecting to the quality of the manuscript. This information is often irrelevant and difficults the understanding of the essential the points of the manuscript.*
- *English can improve. Some expressions doesn't sound natural and there are large sentences that are difficult to read.*
- *The work presents a complexity that is not well developed. I suggest a more plain explanation of certain points. However, if the authors want to explain them with a high level of complexity, it should be better explained.*
- *Re-check on citing.*

We fully agree with your comments regarding the length of the manuscript and that it is necessary to shorten it in order to make it more accessible to readers. In response to your suggestions, we have revised the manuscript, trying to focus on the most important data. Please find all changes in the minor_changes.pdf file.

We have also improved the English language in the manuscript. Expressions that did not sound natural were revised and we broke down the large, complex sentences into smaller, simpler ones to enhance readability.

Moreover, we simplified explanations where possible to enhance overall clarity. For sections where a higher level of complexity is required, we have taken care to provide more detailed and thorough explanations.

The citing styles as well as the order of sources in the reference list were revised.

**Minor suggestions:**

*L20 - with the passage of time the word 'latest' loses meaning.*

The word 'latest' was removed.

*L22 - it seems redundant: 1 decade and approximately 9 years. I'd try to unify these 2 concepts*

The sentence was unified, removing redundant information.

*L25 - concentration instead of content. I think is more appropiate.*

DONE

*L25-26: I think this sentence can be removed: 'Observations indicate an increasing trend in atmospheric CH4 content since 2007, the cause of which is still subject to scientific debate'*

We removed the question. Is the sentence not true anymore? Is the reason for the global increase found in unaccounted anthropogenic emissions?

*L29: to foster understanding OF the global methane cycle*

DONE

*L33: method of choice doesn't sound natural. Maybe... 'typically used methods'*

Reformulated the sentence.

*L43: point sources (Duren, 2019, Nature) instead of point-like sources*

DONE

*L43: just 'This limitation is due...'*

DONE

*L56-58: I don't think references are well written here. Besides, to cite EnMAP mission it is better to cite Guanter, 2015. And probably also PRISMA mission should be cited with another reference.*

Added Guanter et al. (2015) and Cogliati et al. (2021) references.

*L68-69. There is a specification for spectral resolution (¿1nm), but there isn't for spatial resolution.*

In the context of CH4 we consider $<100\,\mathrm{m}$ as high resolution.

*L70-72. Retrievals are the results. Maybe you mean methods to acquire the trace gas retrievals.*

DONE

*L77. Concentration enhancements\**

DONE

*L89 - 'often PRESENT sufficient accuracy'.*

DONE

*L89 - Every nonlinear method are iterative?*

Affirmative, at least in this study.

*L89 - I would reformulate this sentence. Sufficient accuracy seems a good enough accuracy. Then, why nonlinear methods?*

Reformulated "... but often lack accuracy, ..."

*L90 - I don't understand this: 'The retrieval methods are tailored to address the issue of albedo-related biases, which arises due to correlations with broad-band absorption features resulting from the instrument's low spectral resolution.' - I would try a more plain explanation.*

This sentence was removed from the first paragraph in the Methodology section, as it can lead to confusion.

*Figure1-caption - It is difficult to distinguish between the dashed and solid lines. I would try to improve the figure in this aspect. What is 'a' in 'kt/a'? Besides, I think that there is some irrelevant information here for the study (QGIS, flight overpass time-stamps...)*

The 'kt/a' stands for 'kiloton per annum'.

*L96 - conduct or evaluate?*

We removed the sentence to shorten the manuscript.

*L99 . I think the comma ',' can be removed.*

We removed the sentence to shorten the manuscript.

*L100 - I think this is not the best way to cite.*

Please provide an example how we can improve the citation style in this case.

*L106 - What does observation mean? It is not clear. Is important to know that?*

We removed the sentence to shorten the manuscript.

*L106 - '320 across-track detector pixels': this is inherent to the Hyspex SWIR spectrometer measurement.*

We removed the sentence to shorten the manuscript.

*Figure 2-caption - detectors, not pixels. Pixel position?*

Changed detector pixels to detectors.

*Figure 2 - there is a lot to improve in this figure. There is overlapping of numbers, the font from labels makes it difficult to read, the mentioned 'center' is at the left side of the figure, there is two horizontal axis in b (not necessary).*

We revised the figure according to the suggestions.

*L111 - No need for specifying the equivalence in wavenumbers units.*

It is at least needed for the spectral range because we later name the retrieval intervals 4K and 6K, which relates to wavenumbers.

*L115 - not pixel, just detector*

DONE

*L126 - in general?*

Removed "in general".

*L129 - molecular m? not clear*

Clarified this sentence by saying what $m$ stands for.

*L130 - 'Atmosph'erique'*

DONE

*L141 - not pixels*

Detectors

*L145-146 - I'd write 'The highest vertical resolution is found in those layers below...'*

DONE

*Figure3-caption - I don't understand 'mid-infrareddle panel'*

Corrected - it should be "mid-infrared".

*L163 - the new Python version of BIRRA that is used is based on...*

Corrected.

*L170 - 'The transmission by aerosols for different Ångstrom exponents according to is depicted in Fig. 3 (center)' - Incomplete sentence?*

We removed this figure as it is not relevant for the retrieval setups (all non-scattering retrievals).

*L174 - What is 'j'?*

An integer.

*2.3.1. - Here NLS, SLS, and GLS are introduced. But in L159 only NLS and SLS are mentioned.*

The GLS method is not part of the classical/original BIRRA. We added this note to the sentence above.

*L185 - Function 'L' is not defined.*

Removed this function designator as it is not needed.

*L188 - You say 'separable least squares solver' when you already defined SLS.*

DONE

*L198-L200 - How do you get C? Why the location of the point source and wind data must be known?*

In order to not contaminate C with real methane enhancements as it should only account for correlations from the albedo below background atmospheric concentrations (and sensor noise).

*Figure4-caption - Panel show $S^{-1}$, but the caption says: 'backgorund covariance matrix', i.e., C.*

DONE *L209 - Jacobian matrix*

DONE

*Eq 12 and 13 I don't understand where they come from*

Clarified that they were generated by the LLS fit.

*L263 - likelihood*

DONE

*L269 - per pixel doesn't seem possible (maybe per column)*

It should say per measurement.

*L284 - USVt is not explained*

We added the corresponding explanations.

*Figure 5 - Fontsize could improve.*

Increased the figure width.

*L329 - Figure 8b shows*

Figure 8 shows the multi-window fits for scene 09 and 11.

*Figure8-caption - here you shouldn't write 'Best results are acquired for GLS setup'.*

Removed this phrase.

*I don't think the retrieval from averaged data should be showed. They are not so important and can simply be commented in the text.*

The binning of HySpex pixels is performed to reduce noise. Non averaged data is shown in Fig. 9.

*L341 - but the and as?*

We rewrote parts of this paragraph.

*4K and 6K retrievals are shown in some methods and not shown in others. Why so?*

As mentioned in the text, the quality of the linear methods in discriminating the plume from background pixels varies for the spectral intervals (see Table 2 and 3). An example is given in Fig. 12. Figures were chosen to highlight all examined aspects and to be comparable across the different algorithms. However, to show all maps would overload the manuscript (see Fig. 17) and not comply with your request to focus on the most important aspects.

*L344 - to identify*

DONE

*L345 - the lowest*

DONE

*L347-349 - This should not be in this section*

The paragraph was rephrased and shifted to Sec. 3.3

With best regards,

Philipp Hochstaffl and co-authors

---

## Author Response (AR3)

DLR — Remote Sensing Technology Institute
82234 Oberpfaffenhofen
Germany

July 2023

Dear AMT Editorial Board and Reviewers,

first of all we would like to thank again the editor and referees for the constructive and friendly review.

The manuscript has undergone proofread. Only minor changes with respect to the English language have been applied. Apart from these minor changes, the uploaded manuscript is exactly the same as the version of my manuscript accepted by the Associate editor.

With best regards,

Philipp Hochstaffl and co-authors